# Adverse Effect of Sugarcane Extract Powder (SEP) in Hyper-Lipidemic Zebrafish During a 14-Week Diet: A Comparative Analysis of Biochemical and Toxicological Efficacy Between Four SEPs and Genuine Policosanol (Raydel^®^)

**DOI:** 10.3390/ijms26199524

**Published:** 2025-09-29

**Authors:** Kyung-Hyun Cho, Ashutosh Bahuguna, Sang Hyuk Lee, Ji-Eun Kim, Yunki Lee, Cheolmin Jeon, Seung Hee Baek, Krismala Djayanti

**Affiliations:** Raydel Research Institute, Medical Innovation Complex, Daegu 41061, Republic of Korea

**Keywords:** dyslipidemia, fatty liver, oxidative stress, policosanol, senescence, spermatozoa

## Abstract

Sugarcane wax-derived policosanol (POL) is well recognized for its multifaceted biological activities, particularly in dyslipidemia management, whereas sugar cane extract powder (SEP), prepared from whole sugar juice blended with supplementary components, has not been thoroughly investigated for its biological activities and potential toxicities. Herein, the comparative dietary effect of four distinct SEPs (SEP-1 to SEP-4) and Cuban sugarcane wax extracted POL were examined to prevent the pathological events in high-cholesterol diet (HCD)-induced hyperlipidemic zebrafish. Among the SEPs, a 14-week intake of SEP-2 emerged with the least zebrafish survival probability (0.75, log-rank: χ^2^ = 14.1, *p* = 0.015), while the POL supplemented group showed the utmost survival probability. A significant change in body weight and morphometric parameters was observed in the SEP-2 supplemented group compared to the HCD group, while non-significant changes had appeared in POL, SEP-1, SEP-3, and SEP-4 supplemented groups. The HCD elevated total cholesterol (TC) and triglyceride (TG) levels were significantly minimized by the supplementation of POL, SEP-1, and SEP-2. However, an augmented HDL-C level was only noticed in POL-supplemented zebrafish. Likewise, only the POL-supplemented group showed a reduction in blood glucose, malondialdehyde (MDA), AST, and ALT levels, and an elevation in sulfhydryl content, paraoxonase (PON), and ferric ion reduction (FRA) activity. Also, plasma from the POL-supplemented group showed the highest antioxidant activity and protected zebrafish embryos from carboxymethyllysine (CML)-induced toxicity and developmental deformities. POL effectively mitigated HCD-triggered hepatic neutrophil infiltration, steatosis, and the production of interleukin (IL)-6 and inhibited cellular senescence in the kidney and minimized the ROS generation and apoptosis in the brain. Additionally, POL substantially elevated spermatozoa count in the testis and safeguarded ovaries from HCD-generated ROS and senescence. The SEP products (SEP-1, SEP-3, and SEP-4) showed almost non-significant protective effect; however, SEP-2 exhibited an additive effect on the adversity posed by HCD in various organs and biochemical parameters. The multivariate examination, employing principal component analysis (PCA) and hierarchical cluster analysis (HCA), demonstrates the positive impact of POL on the HCD-induced pathological events in zebrafish, which are notably diverse, with the effect mediated by SEPs. The comparative study concludes that POL has a functional superiority over SEPs in mitigating adverse events in hyperlipidemic zebrafish.

## 1. Introduction

Policosanol is a generic name used for the mixture of long-chain aliphatic alcohols (LCAAs) [1] that can be extracted from a variety of sources, including plants (e.g., wheat germ, sugarcane, maize, grapes, rice bran) and animals (e.g., bees) [2,3]. However, policosanol was first extracted from the sugarcane (*Saccharum officinarum* L.) wax originated from Cuba in the early 1990s that harbor a unique mixture of eight LCAAs, namely tetracosanol (C24, 0.1–2%), hexacosanol (C26, 3–10%), heptacosanol (C27, 0.1–3%), octacosanol (C28, 60–70%), nonacosanol (C29, 0.1–2%), triacontanol (C32, 10–15%) and tetratriacontanol (C34, 0.1–5%) by the Centro Nacional de Investigaciones Científicas (CNIC, Havana, Cuba) [1]. Sugarcane is a perennial tropical grass of the Poaceae (Gramineae) family and is mainly cultivated in arid and semi-arid regions for sugar production. Beyond sugar, it serves as a valuable raw material to produce ethanol, vinegar, and wine [4]. Importantly, sugarcane processing yields several byproducts such as bagasse, a lignin-rich residue, and press mud, which is utilized to extract sugarcane wax, a precursor for the extraction of policosanol [5]. Owing to these versatile applications, sugarcane and its derivatives hold significant industrial importance, including in the cosmetic and pharmaceutical sectors [5,6]. Policosanol has been recognized for its diverse functionality, including therapeutic impact on hypertension, intermittent claudication, ischemic heart disease, and Alzheimer’s disease [2,7]. In addition, the inhibitory effect of policosanol has been documented for the synthesis of platelet-aggregating thromboxane B2 (TXB2), consequently attenuating platelet aggregation [7].

Nonetheless, policosanol gained significant attention primarily due to its effect on dyslipidemia [8], by effectively reducing blood total cholesterol (TC), low-density lipoprotein (LDL)-cholesterol, and triglyceride (TG) levels. Additionally, policosanol is recognized for improving the quality and quantity of high-density lipoprotein (HDL) [9], with a marked effect on cholesterol efflux capability [10]. Also, studies confirm policosanol’s effects on protecting LDL oxidation, and the glycation of HDL [11]. Despite this, some studies documented the functional disparity of policosanol towards dyslipidemia [1]. The disparity in policosanol effect may be attributed to several factors, primarily due to the source material as compositional profile, including the types and concentrations of LCAAs of policosanol is strongly dependent on the source material, its geographical origin, and the extraction methodology employed [1,12]. Over 50 distinct policosanol products are currently available in the world market as functional food; however, their functionality has not been not comprehensively evaluated. Nevertheless, some comparative reports indicate considerable differences in their efficacy [13]. Notably, in the Republic of Korea, the Ministry of Food and Drug Safety [MFDS] approved only Cuban sugarcane wax extracted policosanol as a functional food to treat prehypertension [14].

On the other hand, in recent years (2024 onwards), different products formulated with whole sugarcane extract powder (SEP) have appeared in the Korean market as a sugar-processed food from fruits and vegetables. As these products are registered under the general food category, there is no mandatory obligation to disclose the exact composition, unlike the functional foods (like Cuban policosanol), which are strictly regulated in terms of their composition and origin of the source material. Even more, some of these SEP products claimed health-beneficial effects and mimicked functional food, without any substantial scientific evidence.

In view of this, the present study aimed to comparatively evaluate the 14-week dietary intake of four distinct SEP products and policosanol (POL) on the survivability, blood lipoprotein profile, and histopathological changes in vital organs, including liver, kidney, brain, testes, and ovaries, of hyperlipidemic zebrafish. Furthermore, the study investigated the effect of SEPs and POL on plasma oxidative stress and antioxidant variables, such as malondialdehyde (MDA), sulfhydryl groups, ferric ion reduction ability (FRA), and paraoxonase (PON) activity, along with hepatic function biomarkers aspartate aminotransferase (AST) and alanine aminotransferase (ALT). The selection of these markers was based on the fact that MDA represents a key lipid peroxidation product and is widely recognized as an essential indicator of oxidative stress [15], whereas FRA reflects the overall antioxidant capacity of blood [16]. The sulfhydryl groups act as primary antioxidants that scavenge peroxyl radicals [17,18], and their decreased level is linked to diseases like rheumatoid arthritis, coronary heart disease, and kidney disorders [19,20]. PON, an HDL-associated enzyme, exhibits an inverse relationship with lipid peroxidation and oxidative stress [21], and its decreased level is associated with myocardial infarction [22] and chronic liver disease [23]. AST and ALT serve as key biomarkers of liver function, with elevated levels indicating poor liver health.

Zebrafish were chosen as the experimental model due to their high genomic resemblance to humans [24], and more specifically, the close similarity of their lipid metabolism pathways. Zebrafish harbor many key lipid metabolism enzymes, receptors, and lipoproteins analogous to those in humans, including cholesteryl ester transfer protein (CETP), a critical component of human lipid metabolism that is absent in mice [25], rendering zebrafish an ideal model organism for lipoprotein research [26]. Moreover, the high-cholesterol diet (HCD) induced events in zebrafish mimicked with human hepatic steatosis and exhibited gene expression patterns comparable to those observed in mammalian models [27]. Furthermore, HCD-induced responses in zebrafish are parallel to many human pathophysiological conditions, including macrophage lipid accumulation and vascular lesion formation [27]. In addition, zebrafish have proven to be a good model for preclinical research related to human liver [28], kidney [29], and reproductive ailments [30]. Consequently, compounds showing efficacy in zebrafish have a high likelihood of exhibiting comparable responses in human clinical studies.

## 2. Results

### 2.1. Survivability and Alteration in Body Weight

As depicted in the Kaplan–Meier probability curve (Figure 1A), the survivability of zebrafish across the group varied substantially (log-rank: χ^2^ = 14.1, *p* = 0.015). Among all the groups, no changes in the survival probability were reported until week 2. The first decline in survival probability (0.96) was reported in the SEP-1 and SEP-4 supplemented groups at week 4, which remained constant until week 8, and then further declined to 0.93 at week 10, remaining constant thereafter until week 14. A nearly similar trend of the zebrafish survival probability was noticed in the HCD group. on the contrary, no death was reported in the SEP-3 group until week 10; thereafter, survival probability declines slightly (0.96) and remains constant until week 14. The most adverse effect on zebrafish survivability was noticed in the SEP-2 supplemented group, where a severe drop in survival probability (0.86) was noticed at week 8 that progressively declined to 0.78 at week 10 and finally reached 0.75 at week 14. Interestingly, no zebrafish death was noticed in the POL-supplemented group, evident by a perfect constant survival probability (1.0) during the 14-week feeding period.

Further, the survivability analysis of individual male and female zebrafish revealed no gender-based bias in the survivability of zebrafish in HCD, POL, SEP-1, SEP-3, and SEP-4 groups. Unlike this, SEP-2 displayed a male-centric mortality manifested by lower survivability (57.2%) in the male zebrafish than the female zebrafish at week 14 (Appendix A).

The repeated ANOVA outcomes utilizing the multivariate tests (Pillai’s Trace, Wilks Lambda, Hotelling’s Trace, and Roy’s largest root) revealed a significant effect of time (0–14 weeks, F = 148, *p* = 0.00) on the elevation of body weight among all the groups (Figure 1B). Interestingly, all the groups except SEP-2 exhibited nearly identical magnitudes of body weight changes at specific time points that corresponded to the body weight changes observed in the HCD (control) group at the respective time. In contrast, the SEP-2 supplemented group displayed the least body weight gain compared to other groups. Compared to the initial day (week 0) a notable ~2-fold enhancement of the body weight was observed after 14 weeks in the HCD, POL, SEP-1, SEP-3, and SEP-4 supplemented groups. In contrast, after 14 weeks, only a 1.7-fold body weight enhancement with respect to the initial day (week 0) was noticed in the SEP-2 supplemented group (Figure 1B). Notably, at 14 weeks, the SEP-2 supplemented group attained a significantly 18% (*p* < 0.05) lower body weight (506.8 ± 29.8 mg) than the body weight of the HCD consumed group (614.8 ± 27.5 mg), whereas no significant changes in the body weight compared to the HCD group were noticed in rest of the groups (i.e., POL, SEP-1, SEP-3, and SEP-4). Consistently, morphometric analysis revealed a significant change in the body length (BL)/body depth (BD) ratio in the zebrafish from the SEP-2 supplemented groups compared to the HCD group (Figure 1D). Unlike this, non-significant changes in the BL/BD ratio were noticed in POL, SEP-1, SEP-3, and SEP-4 groups with respect to the HCD group.

Intriguingly, the food consumption efficacy examined at week 0 and week 14 showed ~95–100% food consumption across the groups, suggesting that there is no effect of different dietary formulations on liking or disliking and appetite of the zebrafish.

### 2.2. Organ Morphology and Weight

Liver morphology depicted in Figure 2A, revealed hepatomegaly and elevated liver weight in the HCD-consumed (control) group (Figure 2B). The co-supplementation of POL effectively mitigated the HCD-induced hepatomegaly and reduced the liver weight, which was significantly 2.3-fold lower than the liver weight observed in the HCD group. Similar to the POL, co-supplementation of SEP-1 and SEP-2 also mitigates the HCD-induced hepatomegaly and significantly reduces the liver weight. However, when compared with the POL, the liver weight is ~1.5-fold higher in the SEP-1 and SEP-2 supplemented groups. On the contrary, no significant effect of SEP-3 and SEP-4 was observed on the hepatomegaly and liver weight compared to the HCD group.

Examination of kidney morphology and weight revealed nephromegaly and increased kidney weight in the HCD-consumed group, which was significantly prevented by the co-supplementation of POL (Figure 2A,C). The kidney weight in the POL-supplemented groups was significantly 2-fold lower than the kidney weight in the HCD group. No SEP products proved effective in mitigating HCD-induced changes in the kidney weight. Testis morphology and weight revealed no significant change between the HCD, POL, SEP-1, SEP-3, and SEP-4 groups (Figure 2A,E). However, the SEP-2 supplemented group displayed a severe alteration in testis morphology and weight that was significantly 1.8-fold and 1.9-fold inferior compared to the HCD and POL supplemented groups, respectively.

With respect to HCD supplementation, a non-significant effect of POL and SEP products was noticed on the brain and ovary morphology and weight (Figure 2A,D,F).

### 2.3. Blood Lipoprotein Profile and Glucose Levels

An elevated total cholesterol (TC, 198.5 ± 1.5 mg/dL) level was noticed in the HCD-supplemented group (Figure 3A), which is 1.2-fold higher than the basal level (168.3 ± 12.6 mg/dL) detected at week 0 (Appendix A). The co-supplementation of POL (138.1 ± 1.3 mg/dL), SEP-1 (167.4 ± 2.5 mg/dL), and SEP-2 (175.1 ± 1.4 mg/dL) effectively reduced the HCD elevated TC level (Figure 3A). However, when compared with SEP-1 and SEP-2, POL displayed significantly lower TC levels of 18% and 12%, reflecting POL’s higher efficacy over SEP-1 and SEP-2. In contrast, no significant effect of SEP-3 and SEP-4 supplementation was noticed on the HCD-induced TC levels.

Like TC, the basal (week 0) triglycerides (TG, 90.6 ± 6.3 mg/dL) level (Appendix A) was 18% enhanced (106.6 ± 0.4 mg/dL) in the HCD supplemented group which was significantly reduced following the co-supplementation of POL (70.6 ± 2.3 mg/dL), SEP-1 (86.6 ± 3.8 mg/dL), and SEP-2 (88.2 ± 2.6 mg/dL) (Figure 3B). Nonetheless, the POL-supplemented group displayed ~1.2-fold higher efficacy than the SEP-1 and SEP-2-supplemented groups to reducing the HCD-induced TG levels. A non-significant effect of SEP-3 and SEP-4 was noticed to reduce the HCD elevated TG level.

The high-density lipoprotein cholesterol (HDL-C) level (51.5 ± 3.4 mg/dL) at week 0 (Appendix A) was substantially reduced up to 45.9 ± 1.9 mg/dL following 14-week supplementation of HCD. The HCD-depleted HDL-C level was significantly enhanced by 32.4% following the co-supplementation of POL (60.8 ± 1.0 mg/dL) (Figure 3C). Conversely, no effect of any SEP products was observed to elevate the HCD diminished HDL-C level. Even more, in response to SEP-2 supplementation, the HDL-C level (38.7 ± 2.3 mg/dL) was significantly 16% lower compared to the HDL-C level of the HCD group. Likewise, the maximum HDL-C/TC (%) was noticed in the POL supplemented group, which was significantly (*p* < 0.001) higher than the HDL-C/TC (%) quantified in the HCD and SEPs supplemented groups (Figure 3D). Consistently, the HCD elevated TG/HDL-C ratio (2.3) was effectively reversed by the supplementation of POL (1.2) and SEP-1 (1.7) (Figure 3E). Compared to SEP-1, a significantly 29% reduced TG/HDL-C ratio was noticed in the POL-supplemented group. Contrary to SEP-1 and POL, no significant effect of other SEP products was noticed on the reduction of the HCD elevated TG/HDL-C ratio.

The utmost blood glucose level was noticed in the HCD consumed group (125.0 ± 4.5 mg/dL), which was significantly reduced by 1.8-fold (70.5 ± 3.8 mg/dL, *p* < 0.001) following the co-supplementation of POL (Figure 3F). Unlike POL, no significant effect of any SEP product was noticed on the alleviation of HCD-triggered blood glucose level.

### 2.4. Oxidative Variables, Antioxidant Abilities and Hepatic Function Biomarkers of Blood

The plasma malondialdehyde (MDA) level at week 0 (8.2 ± 0.4 μM) (Appendix A) increased up to 10.5 ± 0.5 μM in response to HCD-supplementation for 14 weeks. The HCD elevated MDA level was significantly reduced by 21% following the co-supplementation of POL (8.1 ± 0.3 μM) (Figure 4A). Unlike POL, a non-significant effect of SEP supplementation was noticed on the HCD elevated plasma MDA level. However, SEP-2 exacerbates the HCD-induced MDA level, as evidenced by a 32% higher MDA level in the SEP-2 supplemented group (13.8 ± 0.8 μM) compared to the HCD group.

A substantially compromised plasma sulfhydryl content (11.7 ± 0.5 mmol/mg protein) (Figure 4B) was observed post 14-week intake of HCD with respect to the week 0 level (12.2 ± 0.5 mmol/mg protein) (Appendix A). A significantly 22% higher plasma sulfhydryl content was quantified in the POL supplemented group (14.3 ± 0.6 mmol/mg protein) compared to the HCD group (Figure 4B). Unlike POL, no SEP products showed a significant augmentation of HCD diminished plasma sulfhydryl content. Notably, SEP-2 supplementation exerted a negative impact, resulting in a 23% decrease in plasma sulfhydryl content relative to the HCD group.

Compared to the initial (week 0) FRA (252.9 ± 15.4 μM) and PON ± 0.3 activity (6.9 μU/L/min) (Appendix A), a notable diminished FRA (187.8 ± 5.5 μM) and PON activity (5.8 ± 0.4 μU/L/min) was observed after the 14-week intake of HCD (Figure 4C,D). The HCD-compromised plasma FRA and PON activity were significantly 1.5-fold (285.4 ± 8.6 μM) and 2-fold (11.6 ± 0.5 μU/L/min) enhanced by the co-supplementation of POL (Figure 4C,D). Nevertheless, a non-significant effect of SEP products was noticed on the elevation of plasma FRA and PON activity, which was compromised by the supplementation of HCD. Strikingly, the SEP-2 supplementation displayed a detrimental effect on the plasma FRA and PON activity as reflected by a significantly 1.2-fold and 1.5-fold reduced FRA and PON activity, respectively, relative to the HCD group.

HCD elevated plasma AST and ALT levels were significantly reduced by 29.9% and 26.7% following the co-supplementation of POL. In contrast, no significant (*p* > 0.05) effect of any SEP products was noticed towards the reduction of HCD elevated AST and ALT levels (Figure 4E,F). Furthermore, the SEP-2 supplementation demonstrated a significant 1.2-fold (*p* < 0.05) additive effect toward the HCD elevated AST level.

### 2.5. Effect of Plasma on the CML Induced Toxicity in Zebrafish Embryo

The in vivo functionality assessment of the plasma obtained from different groups was assessed against carboxymethyllysine (CML) induced events in the zebrafish embryos. As depicted in Figure 5A,B, severe embryotoxicity is imposed by CML, where embryo survivability starts to decline 2 h post-injection, and progressively declines with time and finally reaches 11% at 72 h post-injection. Contrary to this, the maximum survivability (~79%) was noticed in the PBS (control) group at 72 h post-injection. Plasma obtained from the different groups showed a substantial counter effect towards CML-induced embryo mortality, as reflected by higher embryo survivability in the groups that received the microinjection of plasma. Among the plasma from the different groups, the lowest survivability was observed in the embryos that received plasma from the SEP-2 group (41%), followed by plasma from the HCD group (48%). While the maximum survivability was noticed in response to plasma from the POL group (71%), this was analogous to embryo survivability as observed in the PBS (control) group. Compared to embryo survivability from HCD and SEP-2 plasma-injected groups, significantly 1.5-fold and 1.7-fold higher embryo survivability were observed in response to the plasma from the POL group.

Further embryo morphological evaluation suggests stunted growth, tail fin curvature (indicated by red arrow), yolk sac edema (indicated by black arrow), pericardial edema (indicated by blue arrow), and reduced somite counts in the majority of embryos from the CML-injected group, depicting a severe teratogenic effect of CML (Figure 5B,C,F). The CML posed severe developmental deformities, substantially countered by the co-injection of plasma from different groups. Among the various groups, the least restrained against CML posed deformities were observed in the embryos injected with plasma from the HCD and SEP-2 groups, where many embryos appeared with tail fin curvature, yolk sac, and pericardial edema with reduced somite counts (15–20). In contrast, embryos injected with plasma from the POL group effectively rescued embryos from the CML-induced developmental deformities as reflected by the heightened somite counts (31) that are analogous to the somite counts observed in the PBS injected group (33).

DHE and AO fluorescent staining revealed a higher extent of reactive oxygen species (ROS) production and apoptosis in the only CML-injected group, which were significantly 9.1-fold and 8.7-fold higher than their basal levels, as observed in the PBS-injected group (Figure 5D,E,G). The microinjection of plasma obtained from the different groups mitigated the CML-triggered ROS generation and apoptosis, as reflected by the significantly (*p* < 0.001) decreased DHE and AO fluorescent intensities. However, when compared between the groups, the best effect was observed in the embryos injected with plasma from the POL group, where a ~5-fold reduced DHE and AO fluorescent intensity was observed compared to the CML injected group. In contrast, the embryos receiving the plasma from the SEP-2 group displayed the least protective effect, as evidenced by a significantly 2.7-fold and 2.5-fold higher DHE and AO fluorescent intensity, respectively, relative to respective intensities observed in embryos injected with the plasma from POL.

### 2.6. Histological Evaluation of Liver

The images of H&E staining, as depicted in Figure 6A,B,E, revealed a heightened neutrophil count in the HCD group, which was significantly reduced by 3.2-fold following HCD co-supplemented with POL. Contrary to the POL, the co-supplementation of all SEPs has no significant effect on reducing the HCD elevated neutrophil counts. Even more, SEP-2 displayed a significantly 1.4 times higher neutrophil count compared to the HCD group.

Consistent with the outcomes of the H&E staining, the IHC staining revealed that POL effectively reduced the HCD-elevated IL-6 level by 2.6-fold (Figure 6C,D,F). In contrast, the co-supplementation of SEP-2 was noticed to augment the HCD-induced IL-6 production, indicated by a significantly 1.8 times higher IL-6 level in the SEP-2 groups with respect to the HCD group. Besides, all the other SEP products failed to produce any significant effect to inhibit the IL-6 level compared to the HCD group.

### 2.7. Fatty Liver Changes, Reactive Oxygen Species Generation and Senescence in Liver

As depicted in Figure 7A,B,D,E, the HCD-consumed group showed a substantial ORO-stained area and DHE fluorescent intensity, highlighting the fatty liver and ROS generation. The supplementation of POL effectively prevents the HCD-induced fatty liver and ROS generation, as indicated by a significant 4-fold and 1.3-fold reduction in ORO-stained area and DHE fluorescent intensity, respectively. Juxtapose, the SEP-1, SEP-2, and SEP-4 supplementation showed a significantly 1.4-fold, 2.6-fold, and 1.4-fold higher ORO-stained area, respectively, than the HCD group, documenting the potential adverse effect of SEP-1, SEP-2, and SEP-4 towards fatty liver. A non-significant effect of SEP-3 was noticed against HCD-induced fatty liver. Similar to the ORO staining, a significantly 1.4-fold (*p* < 0.05) higher DHE fluorescent intensity was logged in the SEP-2 supplemented group than the HCD group. However, a non-significant effect of SEP-1, along with SEP-3 and SEP-4 was observed on HCD-induced ROS production.

The SA-β-gal staining revealed a significantly 6.5-fold and 2-fold lower senescent positive cells in the POL and SEP-1 supplement groups relative to the HCD group (Figure 7C,F). Nevertheless, compared to SEP-1, the POL supplemented group displayed a significantly 3.2 times lower senescent positive cells, attesting to the higher efficacy of POL. In contrast, no significant effect of SEP-3 and SEP-4 was observed towards the inhibition of cellular senescence. However, SEP-2 supplementation emerged with a severe adverse effect, with the highest number of senescent positive cells, which was significantly 1.8-fold and 12-fold higher than that of the HCD and POL supplemented groups.

### 2.8. Histological Analysis of Kidney

The H&E staining revealed sparsely populated distal and proximal tubules with the frequent presence of dilated tubular lumen (indicated by green arrow) and debris in the tubular cast (indicated by red arrow) in the kidney section from the HCD group (Figure 8A). The HCD-induced histological changes are substantially reverted by the co-supplementation of POL. However, the occasional presence of dilated tubular lumen and cellular debris in the tubular cast was noticed. In contrast, the SEP supplementation groups displayed a low protective effect on the renal structure, as reflected by sparsely populated tubular structures with elevated tubular lumen and cellular debris in the tubular cast. However, the most adverse effect was observed in the SEP-2 supplemented groups, where the tubular structures were severely impaired, even the minor presence of a basophilic cluster (dark-purple stained, indicated by the blue arrow) corresponding to new nephron generation was noticed.

The DHE staining revealed a substantial inhibitory effect of POL and SEP-1 supplementation against the HCD-triggered ROS production. Nonetheless, compared to SEP-1, a 1.5-fold decrease in ROS level was quantified in the POL-supplemented group (Figure 8B,D). In contrast to SEP-1, SEP-3 and SEP-4 had no significant effect on reducing ROS production; however, SEP-2 supplementation showed a significantly 12.5% (*p* < 0.05) higher ROS production than the HCD-group, representing the aggravative effect of SEP-2 on the HCD-triggered ROS production.

A high prevalence of senescent positive cells was quantified in the HCD supplemented group, which was significantly reduced by 4.1-fold, 1.4-fold, 1.3-fold, and 1.4-fold following the co-supplementation of POL, SEP-1, SEP-3, and SEP-4 (Figure 8C,E). However, compared with SEP-1, SEP-3, and SEP-4 supplemented groups, a significantly ~3-fold reduced ROS level was observed in the POL supplemented group, underscoring the higher efficacy of POL over SEP products.

### 2.9. Histological Analysis of Brain

The H&E staining revealed no significant histological alterations in the brain sections obtained from the different groups (Figure 9A,B). However, a slightly higher presence of vacuolation and mononuclear cells with clear zones was observed in the tectum optic (TeO) and periventricular gray zone (PGZ) of the brain section from the SEP-1 and SEP-2 supplemented group.

DHE and AO fluorescent staining revealed significantly lower ROS and apoptosis in the brain section of the POL compared to the HCD and SEP supplemented groups (Figure 9C–E,H,I). The SEP-1, SEP-3, and SEP-4 groups showed no significant change in DHE and AO fluorescent intensity compared to the HCD group. Unlike this, the supplementation of SEP-2 displayed a massive ROS generation and apoptosis around PGZ of TeO adjacent to the torus longitudinalis (LS) and lateral division of the vascular cerebelli (Val), which was accounted for 1.4~4.8-fold and 1.8~3.8-fold higher than the ROS and apoptosis detected in the HCD and POL supplemented groups, respectively.

Similarly, the least cellular senescence was observed in the POL-supplemented group, which was significantly 3-fold lower than that in the HCD-consumed group (Figure 9F,G,J). Compared to the HCD group, all SEP-consumed groups displayed a non-significant effect in preventing HCD-induced senescence. Notably, SEP-2 displayed a significantly higher senescence than the HCD consumed group, indicating its provocative effect on HCD-induced senescence in the brain.

### 2.10. Testis Histology

The H&E staining originating from the HCD consumed group showed the elevated space between the seminiferous tubules with a reduced spermatozoa count, which was effectively reverted by the magnitude of 1.4-fold and 1.6-fold by the co-supplementation of POL (Figure 10A,D,E). Contrary to this, no significant effect of SEP-3 and SEP-4 supplementation was observed in preventing HCD-induced changes in the testis histology. Notably, the supplementation of SEP-1 and SEP-2 displayed the aggravative effect on the HCD-induced changes manifested by a significantly 1.3~1.5-fold higher space between the seminiferous tubules and 1.6~1.7-fold reduced spermatozoa counts relative to the HCD consumed group.

The DHE fluorescent staining revealed an effective inhibitory role of POL, SEP-1, SEP-3, and SEP-4 on the HCD-induced ROS generation as depicted by a 2.4-fold, 1.5-fold, 1.4-fold, and 1.4-fold diminished DHE fluorescent intensity with respect to the HCD consumed group, while a non-significant effect was observed in response to SEP-2 supplementation (Figure 10B,F). Compared to SEP-1, SEP-3, and SEP-4, a ~1.7-fold (*p* < 0.01) reduced DHE fluorescent intensity was noticed in the POL-supplemented group, attesting its functional superiority over the SEP products.

The senescent staining suggests a significant 4-fold and 1.4-fold reduction in senescent positive cells in the POL and SEP-1 supplemented groups, respectively, compared to the HCD group (Figure 10C,G). Compared to SEP-1, a significant 2.9-fold reduction in senescent positive cells was observed in the POL-supplemented group, underscoring the higher efficacy of POL over SEP-1 in mitigating the HCD-induced senescence. Besides SEP-1, a non-significant effect of other SEP products was noticed on the inhibition of HCD-guided senescence.

### 2.11. Histological Analysis of Ovaries

The images depicted in Figure 11A,D,E reveal a non-significant difference in the pre- and early oocyte counts in the HCD, POL, or SEP-supplemented groups. In contrast, the HCD diminished mature oocyte counts (2.6%) were significantly enhanced by the co-supplementation of POL (15.6%), while a non-significant effect of SEP-supplemented groups was observed on the enhancement of HCD diminished mature oocytes.

DHE and SA-β-gal staining revealed an effective role of POL, SEP-1, SEP-3, and SEP-4 to reduce the HCD elevated ROS production and cellular senescence (Figure 11B,C,F,G). As depicted in Figure 11F,G, a significant 2.5-fold and 6.6-fold reduction in DHE fluorescent intensity and senescent positive cells was quantified in the POL-supplemented groups, compared to the HCD group. Interestingly, the SEP-1, SEP-3, and SEP-4 supplemented groups displayed nearly similar effects, i.e., a ~1.4-fold reduction in DHE fluorescent intensity and a ~1.8-fold reduction in cellular senescence, respectively, compared to the HCD group. Notably, relative to SEP-1, SEP-3, and SEP-4, a significantly ~1.8-fold (*p* < 0.05) and ~3.5-fold (*p* < 0.001) higher efficiency to inhibit ROS generation and cellular senescence was noticed in the POL-supplemented group.

### 2.12. Multivariate Analysis

The multivariate evaluation using principal component (PCA) and hierarchical cluster analysis (HCA) was conducted to segregate the impact of POL and SEP products from the HCD-induced effect. The test was conducted on data obtained from zebrafish survivability, body weight, blood biochemical and antioxidant variables, and histological outcomes. The PCA scoring plot accounted for 82.9% of the variance and demonstrates a clear separation of HCD-POL and HCD-SEP-2 from the other groups (Figure 12A). The HCD-POL and HCD-SEP were placed in distal negative and positive coordinates of principal component (PC)1, depicting strongly contrasting responses towards the HCD-triggered events.

Furthermore, the HCA supports the PCA findings, placing the HCD and HCD-SEP-4 groups in the same cluster (based on the highest similarity ~93%), and the HCD-SEP-1 and HCD-SEP-3 groups in another cluster (~92% similarity) (Figure 12B). These clusters were further merged and displayed at ~52% similarity with the HCD-SEP-2 group. In contrast, HCD-POL showed the least similarity with these groups. The multivariate analysis clearly depicts a distinct effect of HCD-POL on the SEPs towards the adverse events associated with HCD.

## 3. Discussion

After 14 weeks of supplementation, the survivability of zebrafish in the SEP-2 supplemented group was drastically reduced, indicating a significant adverse effect of SEP-2 on zebrafish survival. Unlike this, better survivability was observed in the other SEP-supplemented groups. Nevertheless, the most notable effect with no zebrafish mortality was observed in the POL-supplemented group, underscoring the safe nature of POL over SEP products, precisely with respect to SEP-2. The high presence of red yeast rice extract (RYR) in the SEP-2 (Table 1) is the primary reason for the high toxicity of SEP-2 towards zebrafish, as RYR has been recognized to be contaminated with a variety of toxic substances that pose a serious health concern [31]. The statement aligns with earlier reports that highlight the severe adverse consequences of RYR consumption [31,32]. The body weight analysis showed no significant effect of POL and SEP-1, SEP-3, and SEP-4 on the HCD-induced elevated body weight of zebrafish. Contrary to this, a significantly reduced zebrafish body weight was observed in the SEP-2-supplemented group, revealing the efficacy of SEP-2 in inhibiting HCD-elevated body weight. The presence of RYR in SEP-2 is the reason for the lower body weight, which is in accordance with previous reports that decipher the anti-obesity effect of RYR [33] by inhibiting the proliferation and differentiation of pre-adipocytes in adipose tissue [34]. Despite the positive impact of SEP-2 on body weight management, severe toxicity (mortality) associated with it poses a safety concern.

HCD consumption has been reported to disrupt the lipoprotein profile, which leads to dyslipidemia in various of model organisms [35,36], including zebrafish [37]. Herein, SEP-1, SEP-2, and POL displayed a substantial effect in mitigating the HCD elevated TC and TG levels. However, compared to SEP-1 and SEP-2 (which contained RYR), POL showed a significantly greater effect in reducing blood TC and TG levels, highlighting the functional superiority of POL over SEP products. The effect of POL has been described to modulate the 3-hydroxy-3-methyl-glutaryl-coenzyme A (HMG-CoA) reductase, a key rate-limiting enzyme of cholesterol biosynthesis, consequently lowering the cholesterol level [1,38]. In addition, the role of hexacosanol (a major LCAA of policosanol) has been noticed to inhibit the nuclear transition of sterol regulatory element-binding proteins (SREBP-2), which is an important transcriptional factor that regulates the expression of HMG-CoA reductase [39]. Additionally, POL stimulates cholesterol catabolism in the liver and facilitates the conversion of cholesterol to bile acids, which are subsequently excreted in the feces, thereby contributing significantly to a decrease in the TC level [3,40]. The cholesterol-lowering effect of SEP-1 and SEP-2 was associated with the presence of RYR in it, which has been recognized to contain monocline K (analogous to lovastatin) [41] that inhibits the activity of HMG-CoA reductase, resulting in a cholesterol-lowering effect [42]. Unlike the effect on TC, all SEP products failed to display an augmenting impact on the HCD diminished HDL-C level. In contrast, POL supplementation effectively elevated the HDL-C level. The findings are in accordance with earlier reports, which represent the positive effect of POL on the augmentation of HDL-C levels in both preclinical [13] and clinical studies [43]. As a primary mechanism, the inhibitory effect of POL on CETP activity has been recognized as crucial in the elevation of HDL-C levels [44]. The current findings align with earlier reports documenting the positive effect of POL in counteracting dyslipidemia in mice and rats [2]. However, no study has demonstrated the effect of SEP on animal models other than zebrafish. The lipid-lowering effect of POL in physiologically distinct species, such as mammals (mice and rats) and vertebrates (zebrafish), testifies to the broad applicability of POL in managing dyslipidemia.

Consistent with the findings of HDL-C, only POL supplementation showed a glucose-lowering effect, while none of the SEP products established a glucose-lowering effect towards the HCD-induced hyperglycemia. The POL glucose-lowering effect is supported by published articles that document the inhibitory impact of POL on blood glucose levels [45]. Notably, high cholesterol contributes substantially to insulin resistance [46], leading to hyperglycemic conditions; fortunately, in addition to cholesterol lowering activity, the POL effect on insulin sensitization and secretion has been well recognized [45], which leads to hypoglycemic effects against HCD and supports the present findings.

MDA is a vital stress marker [47] whose higher levels indicate lipid peroxidation. Likewise, plasma sulfhydryl levels also indicate a stressful environment, and their diminished levels are associated with various diseases [19,20,48]. A positive effect of POL supplementation was noticed towards the HCD-disturbed plasma MDA and sulfhydryl content. Likewise, POL supplementation effectively improved plasma PON and FRA activity, indicating a positive effect of POL on enhancing plasma antioxidant status. The results aligned with previous reports describing the modulatory effect of POL on plasma MDA [44,49], FRA, and PON [9] activity compromised by external stress. Contrary to POL, no SEP products positively modulate the HCD-disturbed plasma MDA, sulfhydryl content, FRA, and PON activity. Even more, in response to SEP-2, the plasma oxidative stress and antioxidant parameters are severely compromised, which were significantly worse than the effect observed in the HCD group, indicating a strong pro-oxidant effect of SEP-2. These results support the survival outcomes, where the SEP-2 group demonstrates markedly reduced survivability. Elevated oxidative stress in the SEP-2 group is likely a key contributor, as oxidative stress is well known to trigger various harmful processes leading to organ damage and cell death [50,51]. More specifically, plasma analysis revealed higher MDA levels in male compared to female zebrafish within the SEP-2 group (Appendix A), which may be a contributing factor to the male-specific mortality. Nevertheless, further detailed investigation is warranted to establish a definitive conclusion. The functionality of plasma obtained from the different groups (i.e., HCD, POL, and SEP products) was tested against CML-induced stress in the zebrafish embryos. The results outlined that the plasma obtained from the POL-supplemented groups exhibited the least ROS production, apoptosis, and a superior embryo-protective role compared to the plasma from the SEP groups. The higher antioxidant activity (as reflected by PON and FRA assays) in the plasma of the POL group is the cause of the least oxidative stress and apoptosis, consequently leading to higher embryo survivability.

HCD and dyslipidemia adversely affect the liver, provoking hepatic inflammation [52,53]. Likewise, high neutrophil counts, fatty liver, and IL-6 production were observed in the HCD group, which was significantly protected by the consumption of POL. The results are in good agreement with established literature, which documents the hepatoprotective role of POL tested in models such as rats [54] and zebrafish [55]. Additionally, hexacosanol (an important LCAA of policosanol) has been recognized to induce autophagy [39], a crucial defensive event in preventing lipid accumulation and fatty liver [39,56]. In contrast to POL, a non-significant effect of SEP products was noticed in mitigating HCD-induced liver damage; nonetheless, the supplementation of SEP-2 has resulted in higher fatty liver changes and inflammation. The presence of RYR in SEP-2 is the core issue underlying the aggravated fatty liver changes and inflammation, which is in accordance with previous reports documenting the adverse effects of RYR on fatty liver and inflammation [32,57]. Consistently, no protective effect of any SEP product was noticed to diminish HCD-triggered ROS generation. Even the SEP-2 supplementation displayed a more pronounced effect, resulting in massive ROS production. In contrast, POL supplementation effectively mitigated the HCD-induced generation of ROS. The cellular antioxidant nature of POL is the primary cause of preventing HCD-induced ROS generation. Consistent with the findings of ROS, a high prevalence of cellular senescence was observed in the HCD group, which intensified further upon supplementation with SEP-2. In contrast, POL supplementation successfully alleviated HCD-induced senescence. The lowest and highest prevalence of cellular senescence in the POL and SEP-2 groups are directly associated with ROS levels in these groups, as ROS-induced oxidative stress has been recognized as a primary instigator of cellular senescence [58,59].

High cholesterol and dyslipidemia are known to adversely affect kidneys [60] and reproductive organs [61,62]. Herein, we have witnessed HCD-induced nephromegaly, kidney damage, elevated ROS, and senescence, which were effectively mitigated by POL supplementation. The positive inhibitory effect of POL on ROS generation manages oxidative stress, which is one of the leading causes of kidney damage [63]. Unlike POL, SEP-2 displayed a non-significant protective effect against HCD-induced kidney damage. The presence of RYR in SEP-2 product is likely responsible for kidney damage, as several reports have documented the kidney-damaging effect of RYR [64,65].

Similar to its protective effect on the liver and kidney, POL supplementation effectively reduces ROS generation, apoptosis, and senescence in the brain, resulting in improved brain health compared to the HCD group. In contrast to POL, SEP-2 displayed an additive effect on the HCD-induced adverse events in the brain. The brain-protective effect of POL is linked to its substantial impact on liver and kidney health. The notion aligns with reports that illustrate the liver–brain [66] and kidney–brain axis [67,68]. Accumulated fat in the liver altered the composition of lipid-derived products in circulation, which modulates the blood–brain barrier permeability and facilitates the accumulation of toxic substances and inflammatory cells in the brain, leading to brain damage [69]. Likewise, oxidative stress in chronic kidney disease contributes to brain lesions, and cognitive decline leads to neuropsychiatric disorders, underlying an axis between the kidney and the brain [67]. Unlike POL, SEP-2 damages the liver and kidneys, thus having an adverse effect on the brain through these interrelated pathways.

In addition, a substantially better protective effect of POL was noticed on testis and ovary health compared to SEP products. Precisely when compared with SEP-2, POL showed significant inhibition of ROS and senescence in the testis and ovaries. Studies concerning the POL effect on reproductive organs are limited; however, a preliminary study noted a positive impact of POL on the egg-laying behavior and survivability of zebrafish eggs [55]. It is apparent that the cellular antioxidant nature of POL substantially protects the ovary and testis. The perspective is supported by reports documenting the positive influence of antioxidants in protecting the ovary [70] and testis [71] against external stimuli. Unlike POL, SEP-2 exhibited a compromised antioxidant effect, as evidenced by the high prevalence of ROS in the ovaries and testis, and thus failed to protect the reproductive organs.

## 4. Materials and Methods

### 4.1. Materials

Policosanol (abbreviated as POL batch no: 2325), a mixture of eight long aliphatic alcohols (LCAA, C24 to C34), was extracted from sugarcane wax sourced from Cuba, and was complementarily provided by Raydel^®^ Pty. Ltd., Thornleigh, NSW, Australia. Four different sugarcane extract powders (SEP), namely JW Pharmaceuticals, Gwacheon, Republic of Korea (abbreviated as: SEP-1, batch: 1183001), Nutricore, Seoul, Republic of Korea (abbreviated as: SEP-2, batch no: 2400029), Esther formula, Seoul, Republic of Korea (abbreviated as: SEP-3, batch no: 2400561), and MayjuneNutri’s, Seoul, Republic of Korea (abbreviated as: SEP-4, batch no: 2983411), were purchased from the local market, Daegu, Republic of Korea. A specification of POL and SEP-1 to SEP-4 is provided in Table 1. All the other chemicals and reagents were used as supplied unless otherwise stated.

### 4.2. Zebrafish Culturing

Twelve-week-old zebrafish (AB strain, n = 200) of mixed sex [i.e., male (n = 100) and female (n = 100)] were housed in the aeriated tank equipped with a constant rotating water supply. Zebrafish were maintained at 28 °C under a 14 h light and 10 h dark photo period, adhering to the guidelines of Animal Use and Care adopted by Raydel Research Institute (approval code RRI-23-007, 27 July 2023, date of approval). A normal tetrabit flakes (ND, Tetrabit Gmbh D49304, Melle, Germany) was fed to the zebrafish twice a day. Zebrafish were acclimatized to the laboratory conditions (for 2 weeks) prior to starting the experiment.

### 4.3. Formulation of Different Diets

Normal tetrabit (ND) was amalgamated with cholesterol (4% *w*/*w*) to make the high-cholesterol diet (HCD). In brief, 750 g of ND was suspended in 30 g of cholesterol (i.e., final 4% *w*/*w*) and mixed thoroughly using a spatula. Subsequently, chloroform was added, and the mixture was vigorously agitated to distribute the cholesterol evenly. Finally, the chloroform was evaporated entirely in the fume hood, and the HCD was further used to prepare different POL and SEPs formulated diets. Briefly, HCD (100 g) was mixed either with 1% (*w*/*w*) POL or SEP-1/SEP-2/SEP-3/SEP-4 to make five different dietary formulations named HCD+POL, HCD+SEP-1/SEP-2/SEP-3 and SEP-4, respectively, (as depicted in Table 1). Different diets, i.e., HCD and HCD supplemented either with POL or SEPs, were preserved in a refrigerator (4 °C) and used to feed the zebrafish for 14 weeks. After using it for feeding (every day), different diets were transferred to the refrigerator.

### 4.4. Feeding of Different Diets to Zebrafish

A total of 168 zebrafish were allocated into six cohorts (n = 28/group), with each group comprising an equal number of male (n = 14) and female (n = 14) zebrafish that were kept in separate tanks. The groups were assigned the following diets: HCD alone (group I), HCD+POL (group II), HCD+SEP-1 (group III), HCD+SEP-2 (group IV), HCD+SEP-3 (group V), and HCD+SEP-4 (group VI). Within each group, zebrafish (n = 28, 14 male and 14 female) were further distributed into 4 separate tanks (i.e., n = 7/tank, 2 tanks containing male and 2 tanks containing female) and provided with the specified diet of 70 mg/tank two times in the day (~9 a.m. and 6 p.m.), amounting to a total of 140 mg/tank/day (equivalent to ~20 mg/zebrafish/day). Prior to the specified dietary intervention, all zebrafish in groups I-VI were pre-fed exclusively with HCD for 7 weeks to induce metabolic stress (hyperlipidemic zebrafish), after which they were maintained on their designated diets. Notably, at the end of 7 weeks of pre-feeding, blood was collected from ten randomly selected zebrafish and analyzed for biochemical analysis to ensure the induction of metabolic stress. Appendix A depicts the change in the blood-biochemical profile after 7 weeks of pre-feeding with HCD compared to their initial blood-biochemical profile while consuming the ND diet.

Food consumption among all groups was examined using the previously described method for zebrafish [72]. The food consumption was determined at the beginning (week 0) and at the end of 14 weeks using the following formula: [(total food given per group–residual food per group)/total food given per group] × 100. The residual food was quantified post 30 min of the feed supplied.

### 4.5. Assessment of Zebrafish Survivability, and Body Weight

Zebrafish across groups (I–VI) were monitored every day until the end of the 14 weeks to examine their survivability. Also, within the group, the survivability of male and female zebrafish was recorded separately. The body weight of zebrafish was quantified gravimetrically at the interval of 2 weeks, starting from week 0 to week 14. For the body weight analysis, zebrafish were anesthetized by drenching them in the 2-phenoxyethanol (0.1%) solution for 1 min. The anesthetized zebrafish were kept on tissue paper, and their body weight was immediately measured using an electronic balance (Ohaus, Parsippany-Troy Hills, NJ, USA).

### 4.6. Blood and Organ Collection

Post 14 weeks, zebrafish in each group were starved overnight before collecting the blood. For the blood collection, starved zebrafish were scarified using hypothermic shock, and blood (~2–5 μL/zebrafish) was immediately collected and mixed with phosphate-buffered saline (PBS)-ethylenediaminetetraacetic acid (EDTA 1 mM) in 2:3 (*v*/*v*). Blood samples were collected from all groups on the same day, between 9 a.m. to 10.30 a.m. to minimize circadian and feeding-related variabilities.

Liver, kidney, brain, testis, and ovaries were removed surgically and preserved separately in the formaldehyde solution (10%) for further histological uses.

### 4.7. Blood Biochemical Analysis

The plasma lipid levels concerning total cholesterol (TC), triglycerides (TG), high-density lipoprotein cholesterol (HDL-C), and plasma liver function markers aspartate aminotransferase (AST) and alanine aminotransferase (ALT), were quantified using the commercial kits following the guidelines of the manufacturers. The plasma malondialdehyde (MDA) and ferric reducing ability (FRA) were quantified using the earlier adopted method [9]. A detailed methodology for quantifying plasma lipid profile, hepatic function markers, MDA, and FRA is provided in the Appendix A.

Plasma sulfhydryl content and paraoxonase (PON) activity were determined using the earlier described method [73]. For the quantification of sulfhydryl content, an equal volume of plasma (50 μL, 1 mg/mL protein) was mixed with 0.4% of 5,5-dithio-bis-(2-nitrobenzoic acid) and content was incubated at room temperature (RT) for 2 h. Subsequently, absorbance at 412 nm was recorded, and the sulfhydryl content was quantified as mmol/mg protein using the molar absorbance coefficient (ε) 1.36 × 10^4^ M^−1^cm^−1^ of formed product (5-thiol-2-nitrobenzoic acid).

For the PON activity, plasma (40 μL, 1 mg/mL protein) was mixed with paraoxon ethyl (0.15 g/mL, 160 μL), and content was incubated for 2 h at room temperature, and the absorbance at 415 nm was recorded. The PON activity was quantified using the molar absorbance coefficient (ε) 1.7 × 10^3^ M^−1^cm^−1^ of product (*p*-nitrophenol).

An automated digital glucose meter (AccuCheck, Roche, Basel, Switzerland) was used for the quantification of the blood glucose level.

### 4.8. In Vivo Functionality Evaluation of the Plasma

The functionality of the plasma obtained from the different groups (I–VI) was analyzed in zebrafish embryos against carboxymethyllysine (CML) triggered toxicity, using an earlier described method [73].

#### 4.8.1. Embryo Collection and Microinjection of Plasma

For embryo production, adult male and female zebrafish were kept in the breeding tank and segregated from each other for ~16 h using a physical divider. Subsequently, the divider was removed, allowing the zebrafish to mate uninterrupted for ~30 min. The produced embryos were collected, rinsed with water, and then kept in the 3% sea salt solution (containing 1 μg/mL methylene blue).

The collected embryos (1.5 h post-fertilization) were divided into eight different groups (n = 150/group). Embryos were microinjected with 10 nL PBS (group I, control). A 500 ng CML dissolved in 10 nL PBS was injected in the embryos (group II), while the 500 ng CML suspended in the 10 nL of plasma (equivalent protein 1 mg/mL) obtained either from HCD alone (group III), HCD+POL (group IV), HCD+SEP-1 (group V), HCD+SEP-2 (group VI), HCD+SEP-3 (group VII), and HCD+SEP-4 (group VIII) was microinjected in the embryos of the respective groups. Microinjection among all the embryos was made under the microscope using a microcapillary pipette equipped with a pneumatic picopump (PV830, World Precision Instruments, Sarasota, FL, USA). Notably, among all the groups, microinjection was made in a nearly similar position of the yolk to minimize the bias. The treated embryos were visualized under the microscope for 72 h post-injection to examine the survivability and developmental deformities following the OECD 2019 guidelines [74].

#### 4.8.2. Staining for the Reactive Oxygen Species (ROS) and Apoptosis in Embryos

The generation of reactive oxygen species (ROS) and apoptosis in the embryos were examined by the dihydroethidium (DHE) and acridine orange (AO) staining [75]. Briefly, 10 embryos from each group were placed in a 24-well culture plate and incubated with 500 μL of a DHE (30 μM) and AO (5 μg/mL) solution for 30 min in the dark. The stained embryos were rinsed three times with 1× PBS and visualized under a fluorescent microscope at the excitation and emission wavelengths of 585 nm/615 nm (for DHE) and 505 nm/535 nm (for AO).

### 4.9. Histological Analysis

For the histological analysis, the whole organ (liver, kidney, brain, testis, and ovaries) was blocked in the FSC22 frozen solution (Leica, Nussloch, Germany). Notably, the respective organs from different groups were inserted in the FSC22 frozen solution in the same orientation to prepare the solid block, orienting nearly similar anatomical regions for sectioning. The fixed tissue block was sectioned (7 μm thick) using the cryo-microtome (Leica CM-1510S, Nussloch, Germany). The sections obtained from the different tissues were processed for the hematoxylin and eosin (H&E) staining using the previously described method [76].

For the oil red O staining [75], the liver tissue section (7 μm) was covered with 500 μL of ORO solution (0.1%) and incubated for 5 min at 60 °C. The stained section was washed three times with isopropanol (60%), dried, and visualized microscopically.

### 4.10. Immunohistochemical (IHC), Cellular Senescence and Fluorescent Staining

The liver section (7 μm) was processed for the IHC staining to detect the interleukin (IL)-6 production [77]. Briefly, the tissue section was covered for 16 h with the IL-6-specific immunoglobulins (200× diluted, Abcam ab9324, Cambridge, UK) in the cool, moist environment. Following the development of the section using the EnVision+system HRP polymer kit (codeK4001, Dako, Glostrup, Denmark) containing HRP-tagged secondary immunoglobulin (1000× diluted). The developed IHC section was visualized under the microscope. To enhance the clarity, the IHC-stained images were processed for the red conversion at the brown color threshold value of 20–120 employing the ImageJ software (https://imagej.net/ij, assessed on 6 June 2025, 1.53 version).

Senescent-associated β-galactosidase staining (SA-β-gal) was performed to detect the cellular senescence [13]. In brief, the tissue section was covered for 16 h with 5-bromo-4-choloro-3-indolyl-β-D-galactopyranoside solution (X-gal, 0.1%). The section was washed with tap water and visualized under the microscope for the detection of blue-stained senescent positive cells.

DHE and AO fluorescent staining were performed using the methodology described in Section 2.3.

### 4.11. Stastistical Analysis

Statistical significance between the groups was examined using one-way analysis of variance (ANOVA) followed by Tukey’s post hoc analysis employing SPSS software (version 29, Chicago, IL, USA). The data normal distribution was assessed prior to performing the one-way ANOVA. The multivariate examination with respect to principal component analysis (PCA) and hierarchical cluster analysis (HCA) was used using Minitab Statistical software (version 21.4, State College, PA, USA).

## 5. Conclusions

A 14-week dietary intervention study showed a severe adverse effect of SEP-2 (containing RYR) on the survivability of hyperlipidemic zebrafish, while no visible adverse effect was noted for POL. In contrast to the POL, SEP-2 elevated oxidative stress and diminished antioxidant variables of plasma. Similarly, SEP-2 exacerbates the HCD-induced fatty liver, hepatic inflammation, damage to the kidney, brain, and reproductive organs, while POL displays a notable protective effect against the HCD-induced adverse events. The other SEP products (SEP-1, SEP-3, and SEP-4) displayed no improved effect against HCD-induced dyslipidemia. The antioxidant and anti-inflammatory effects of POL contributed to its protective effect against HCD-induced dyslipidemia and oxidative stress. Whereas SEP-2 displayed a pro-oxidant effect, consequently aggravating the HCD-induced adverse events, leading to severe toxicity. The study concludes the functional divergence between POL and SEP products and underscores the specific role of POL in alleviating HCD-mediated adverse outcomes. Nevertheless, a limited number of zebrafish per group and the unavailability of exact compositional details of SEPs emerged as a fundamental limitation of the study that needs to be addressed in the future.

## Figures and Tables

**Figure 1 ijms-26-09524-f001:**
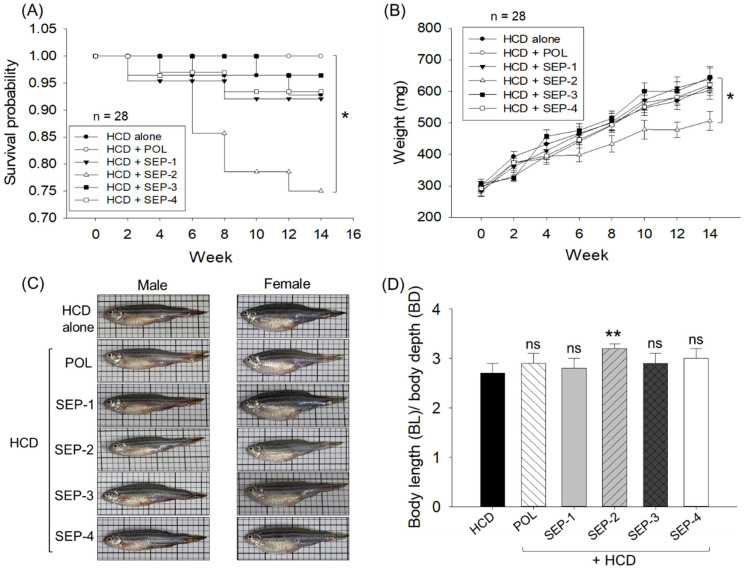
Comparative effects of policosanol and different sugarcane extract powders on the survivability and body weight of hyperlipidemic zebrafish. (**A**) Kaplan–Meier survival probability curve during 0–14 weeks. The * highlights the statistical difference (*p* < 0.05, log-rank: χ^2^ = 14.1). (**B**) Body weight change over 14 weeks; * and ** represent the statistical difference at *p* < 0.05 and *p* < 0.01, respectively, compared to the HCD group. (**C**) Representative images of zebrafish captured post 14 weeks of dietary intake. (**D**) Morphometric analysis of the zebrafish [expressed as the ratio of body length (BL) and body depth (BD)]. The acronyms HCD: High cholesterol diet, POL: policosanol, and SEP: sugarcane extract powder.

**Figure 2 ijms-26-09524-f002:**
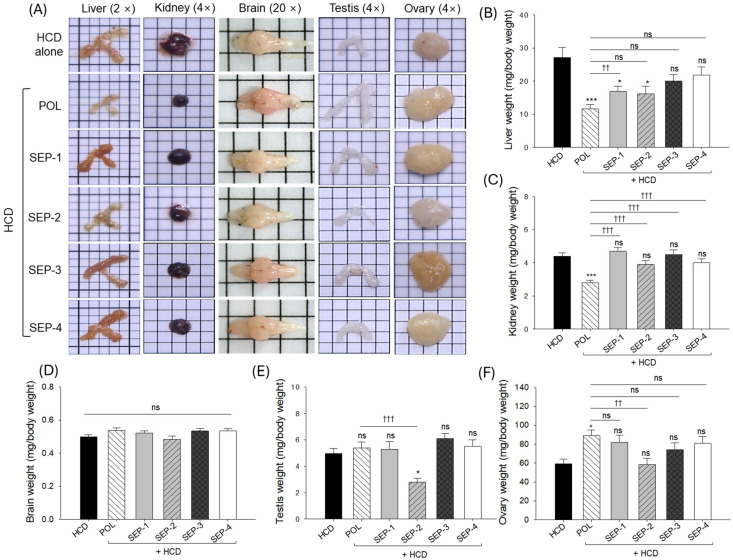
Organ morphology and organ weight of hyperlipidemic zebrafish post 14 weeks consumption of policosanol and different sugarcane extract powder. (**A**) Representative organ (liver, kidney, brain, testis, and ovary) morphology obtained from different groups. Average weight of (**B**) liver, (**C**) kidney, (**D**) brain, (**E**) testis, and (**F**) ovary. The * and *** represent the statistical difference at *p* < 0.05 and *p* < 0.001 with respect to the HCD group, while ^††^ (*p* < 0.01) and ^†††^ (*p* < 0.01) highlight the statistical significance vs. the POL group. The “ns” represents the non-significant difference. The acronyms HCD: High cholesterol diet, POL: policosanol, and SEP: sugarcane extract powder.

**Figure 3 ijms-26-09524-f003:**
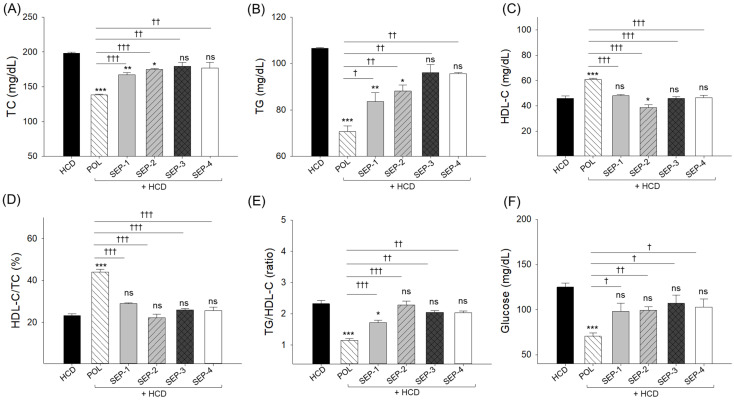
A comparative effect of 14 weeks of dietary intake of policosanol and different sugarcane extract powder on the hyperlipidemic zebrafish blood (**A**) total cholesterol (TC), (**B**) triglycerides (TG), (**C**) high-density lipoprotein cholesterol (HDL-C), (**D**) percentage ratio of HDL-C/TC, (**E**) TG/HDL-C ratio and (**F**) glucose levels. The *, ** and *** represent the statistical difference at *p* < 0.05, *p* < 0.01, and *p* < 0.001 with respect to the HCD group, while ^†^ (*p* < 0.05), ^††^ (*p* < 0.01), and ^†††^ (*p* < 0.001) highlight the statistical significance vs. the POL group. The “ns” represents the non-significant difference. The acronyms HCD: High cholesterol diet, POL: policosanol, and SEP: sugarcane extract powder.

**Figure 4 ijms-26-09524-f004:**
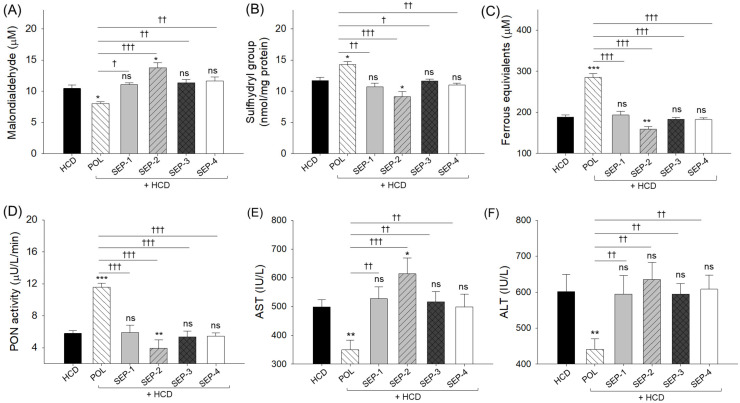
A comparative effect of 14 weeks of dietary intake of policosanol and different sugarcane extract powders on the hyperlipidemic zebrafish-blood (**A**) malondialdehyde (MDA) level, (**B**) sulfhydryl content, (**C**) ferric ion equivalent, (**D**) paraoxonase (PON) activity, (**E**) aspartate aminotransferase (AST), and (**F**) alanine aminotransferase (ALT) level. The *, ** and *** represent the statistical difference at *p* < 0.05, *p* < 0.01, and *p* < 0.001 with respect to the HCD group, while ^†^ (*p* < 0.05), ^††^ (*p* < 0.01), and ^†††^ (*p* < 0.001) highlight the statistical significance vs. the POL group. The ns represents the non-significant difference. The acronyms HCD: High cholesterol diet, POL: policosanol, and SEP: sugarcane extract powder.

**Figure 5 ijms-26-09524-f005:**
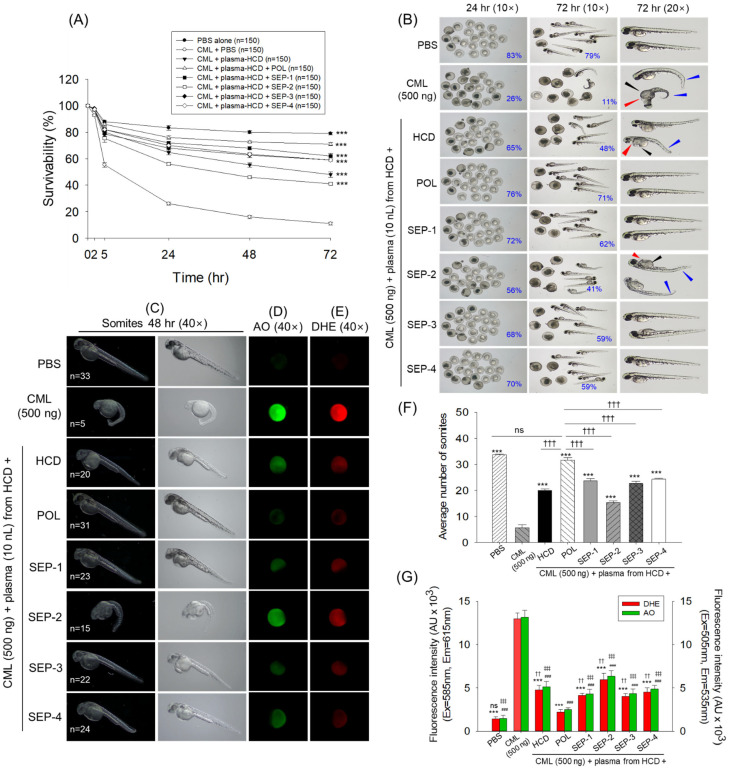
Effect of plasma obtained from the 14-week policosanol and different sugarcane extract powder-supplemented zebrafish on the carboxymethyllysine-induced toxicity in zebrafish embryos. (**A**) Embryos’ survivability kinetics (0–72 h post-injection) across different groups. (**B**) Pictorial view of the zebrafish embryos at 24 h and 72 h post-injection of plasma obtained from the different groups; red arrow depicts tail fin curvature, black and blue arrows depict yolk sac edema and pericardial edema, respectively. Numbers in the blue font represent the survivability of embryos. (**C**) Representative images indicating somite numbers in the developing embryos at 48 h post-injection. Numbers in the white font represent the average somite counts. (**D**,**E**) Dihydroethidium (DHE), and acridine orange (AO) fluorescent staining. (**F**) Average somite numbers in the embryos. (**G**) Quantification of the DHE and AO fluorescent intensities using ImageJ software (1.53 version, https://imagej.net/ij, assessed on 6 June 2025). The *** (*p* < 0.001) represent the statistical difference (for the data of survivability, somite count, and DHE fluorescent intensity), whereas ^###^ (*p* < 0.001) represents the statistical difference for AO fluorescent intensity with respect to the CML+PBS group, respectively. The sign ^††^ (*p* < 0.01), and ^†††^ (*p* < 0.001) represent the statistical difference (for the data of somite counts and DHE fluorescent intensity), while ^‡‡‡^ (*p* < 0.001) highlights the statistical significance for the AO fluorescent vs. POL group. The ns represents the non-significant difference. The acronyms CML: carboxymethyllysine, PBS: phosphate-buffered saline, HCD: High cholesterol diet, POL: policosanol, and SEP: sugarcane extract powder.

**Figure 6 ijms-26-09524-f006:**
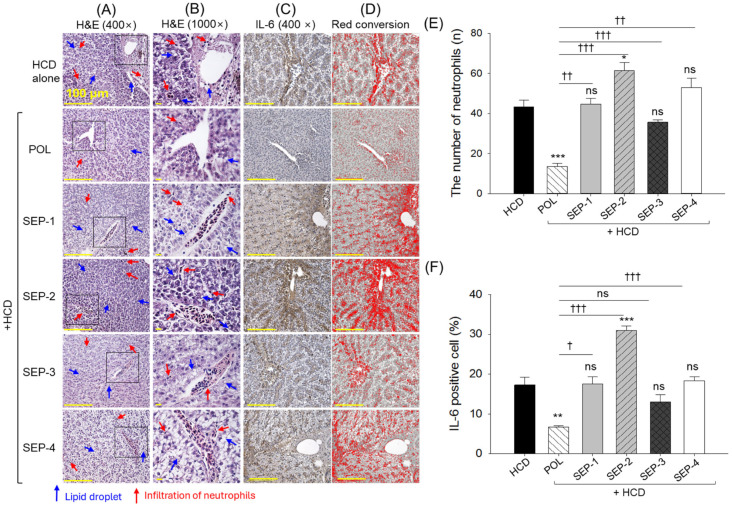
Histological analysis of liver from the hyperlipidemic zebrafish post 14 weeks of dietary intake of policosanol and different sugarcane extract powder. (**A**) Hematoxylin and eosin (H&E) staining, (**B**) magnified view of the H&E-stained section covered under the black dotted box, (**C**) immunohistochemical staining (IHC) for the detection of interleukin (IL)-6, (**D**) red conversion of the IHC-stained area to improve visibility. Red conversion was carried out using Image J software (1.53 version, https://imagej.net/ij, assessed on 6 June 2025) at the brown color threshold value (20–120). (**E**,**F**) Quantification of the number of neutrophils and IL-6-stained area, respectively. The *, **, and *** represent the statistical difference at *p* < 0.05, *p* < 0.01, and *p* < 0.001 with respect to the HCD group, while ^†^ (*p* < 0.05), ^††^ (*p* < 0.01), and ^†††^ (*p* < 0.001) highlight the statistical significance vs. the POL group. The ns represents the non-significant difference. The acronyms HCD: High cholesterol diet, POL: policosanol, and SEP: sugarcane extract powder.

**Figure 7 ijms-26-09524-f007:**
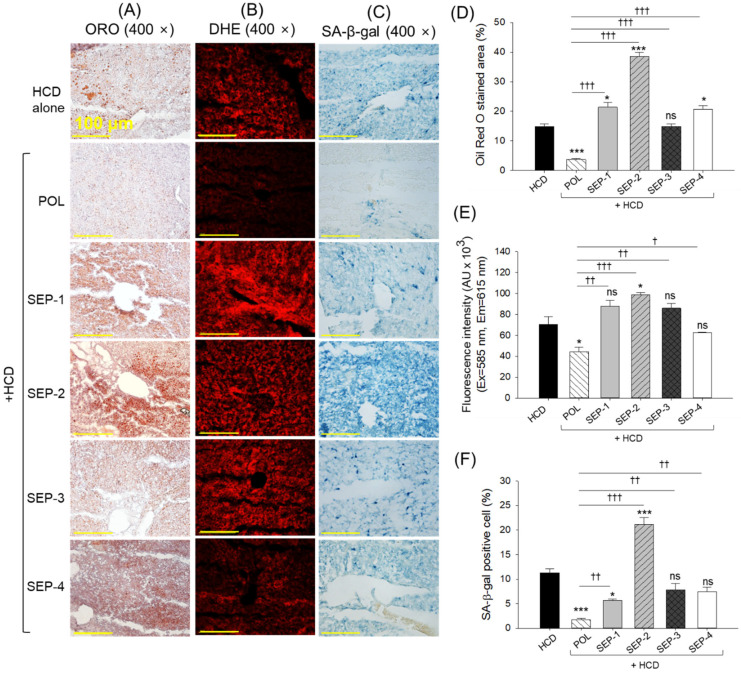
(**A**) Oil red O (ORO) staining, (**B**) dihydroethidium (DHE) staining, and (**C**) senescent-associated β-galactosidase (SA-β-gal) staining of liver from the hyperlipidemic zebrafish post 14 weeks dietary intake of policosanol and different sugarcane extract powder [scale bar, 100 μm]. (**D**–**F**) Quantification of ORO-stained area, DHE fluorescent intensity, and senescent positive cells, respectively. The * and *** represent the statistical difference at *p* < 0.05 and *p* < 0.001 with respect to the HCD group, while ^†^ (*p* < 0.05), ^††^ (*p* < 0.01), and ^†††^ (*p* < 0.001) highlight the statistical significance vs. the POL group. The ns represents the non-significant difference. The acronyms HCD: High cholesterol diet, POL: policosanol, and SEP: sugarcane extract powder.

**Figure 8 ijms-26-09524-f008:**
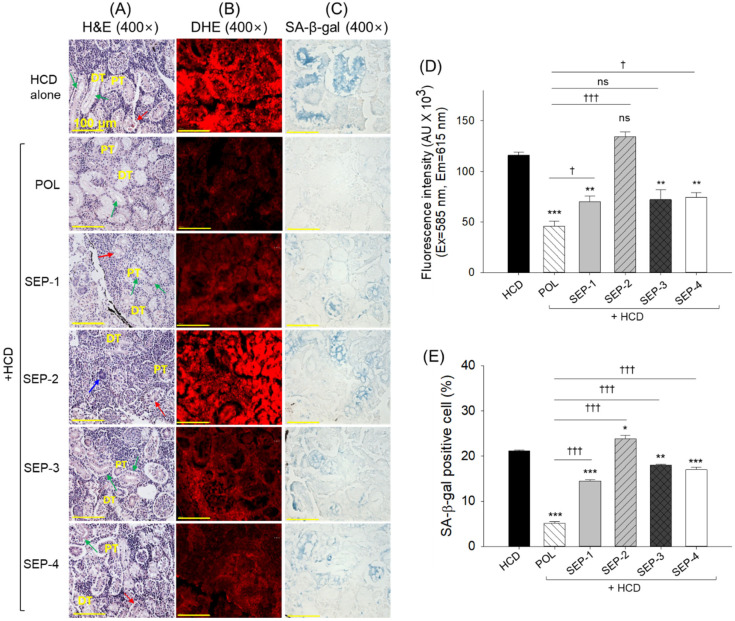
Histological analysis of the kidney from the hyperlipidemic zebrafish post 14 weeks of dietary intake of policosanol and sugarcane extract powder. (**A**) Hematoxylin and eosin (H&E) staining; green arrow indicates dilated tubular lumen, red arrow indicates debris in the tubular cast, and blue arrow represents basophilic cluster corresponding to new nephron generation. (**B**) dihydroethidium (DHE) fluorescent staining, (**C**) senescent-associated β-galactosidase (SA-β-gal) staining [scale bar, 100 μm]. (**D**,**E**) quantification of DHE fluorescent intensity, and SA-β-gal positive cells, respectively. The *, **, and *** represent the statistical difference at *p* < 0.05, *p* < 0.01, and *p* < 0.001 with respect to the HCD group, while ^†^ (*p* < 0.05), and ^†††^ (*p* < 0.001) highlight the statistical significance vs. the POL group. The “ns” represents the non-significant difference. The acronyms HCD: high-cholesterol diet, POL: policosanol, and SEP: sugarcane extract powder.

**Figure 9 ijms-26-09524-f009:**
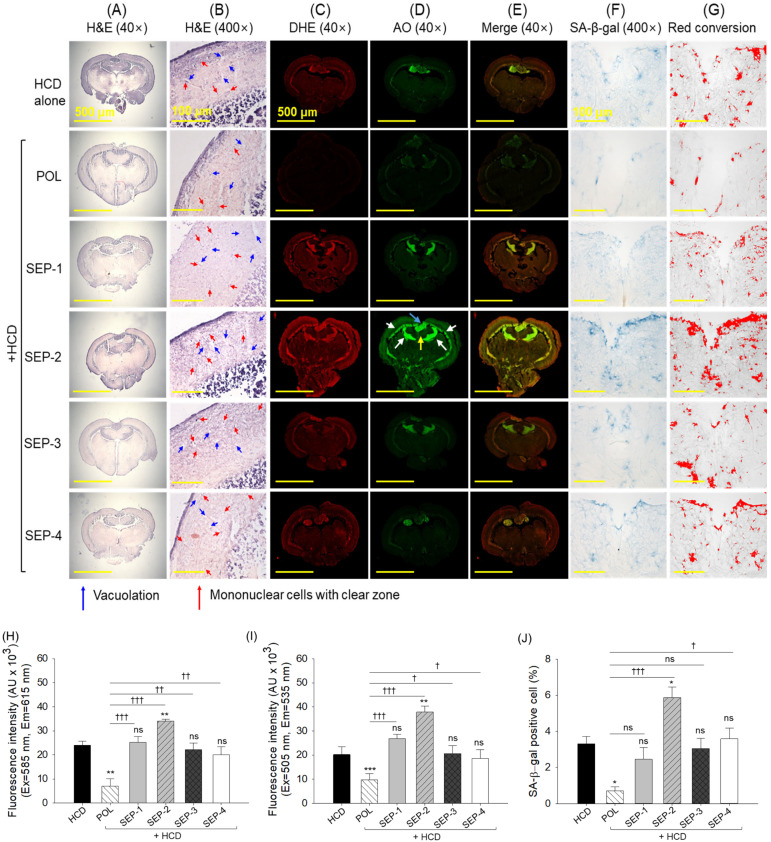
Brain histology of hyperlipidemic zebrafish post 14 weeks dietary intake of policosanol and different sugarcane extract powder. (**A**,**B**) Hematoxylin and eosin (H&E) staining at 40× and 400× magnification. (**C**–**E**) Dihydroethidium (DHE), acridine orange (AO), and merged images of DHE and AO fluorescent staining, respectively. The white arrows represent the periventricular grey zone (PGZ) of tectum opticum (TeO), the blue arrow highlights the torus longitudinalis (LT), and the yellow arrow highlights the lateral division of the vascular cerebelli (Val). (**F**) Senescent-associated β-galactosidase (SA-β-gal) staining. (**G**) Image J-based red conversion of the SA-β-gal-stained area (at the blue color threshold value 0–120) to enhance the visibility. (**H**,**I**) Quantification of DHE and AO fluorescent intensity, respectively. (**J**) Quantification of SA-β-gal positive cells. The * and ** represent the statistical difference at *p* < 0.05, and *p* < 0.001 with respect to the HCD group, while ^†^ (*p* < 0.05), ^††^ (*p* < 0.01) and ^†††^ (*p* < 0.001) highlight the statistical significance vs. the POL group. The ns represents the non-significant difference. The acronyms HCD: high-cholesterol diet, POL: policosanol, and SEP: sugarcane extract powder.

**Figure 10 ijms-26-09524-f010:**
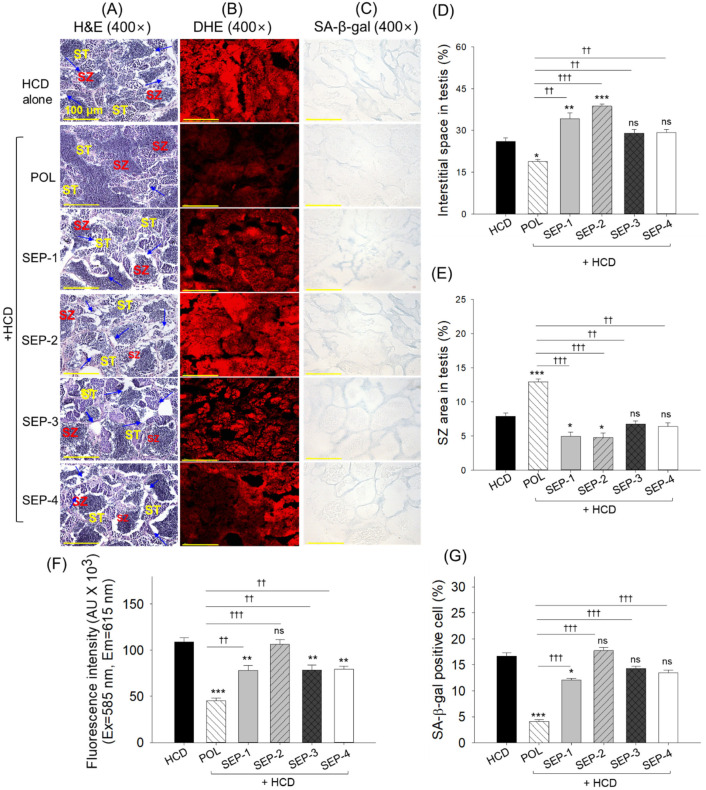
Histological analysis of testis from the hyperlipidemic zebrafish post 14 weeks of dietary intake of policosanol and different sugarcane extract powder. (**A**) Hematoxylin and eosin (H&E) staining, (ST: spermatocytes, SZ: spermatozoa), blue arrow highlights disruption of the basal lamina membrane. (**B**) Dihydroethidium (DHE) fluorescent staining. (**C**) Senescent-associated β-galactosidase (SA-β-gal) staining [scale bar, 100 μm]. (**D**,**E**) Quantification of the interstitial space between the seminiferous tubules and spermatozoa counts, respectively. (**F**,**G**) Quantification of DHE fluorescent intensity and SA-β-gal positive cells, respectively. The *, **, and *** represent the statistical difference at *p* < 0.05, *p* < 0.01, and *p* < 0.001 with respect to the HCD group, while ^††^ (*p* < 0.001) and ^†††^ (*p* < 0.001) highlight the statistical significance vs. the POL group. The ns represents the non-significant difference—the acronyms HCD: high-cholesterol diet, POL: policosanol, while SEP: sugarcane extract powder.

**Figure 11 ijms-26-09524-f011:**
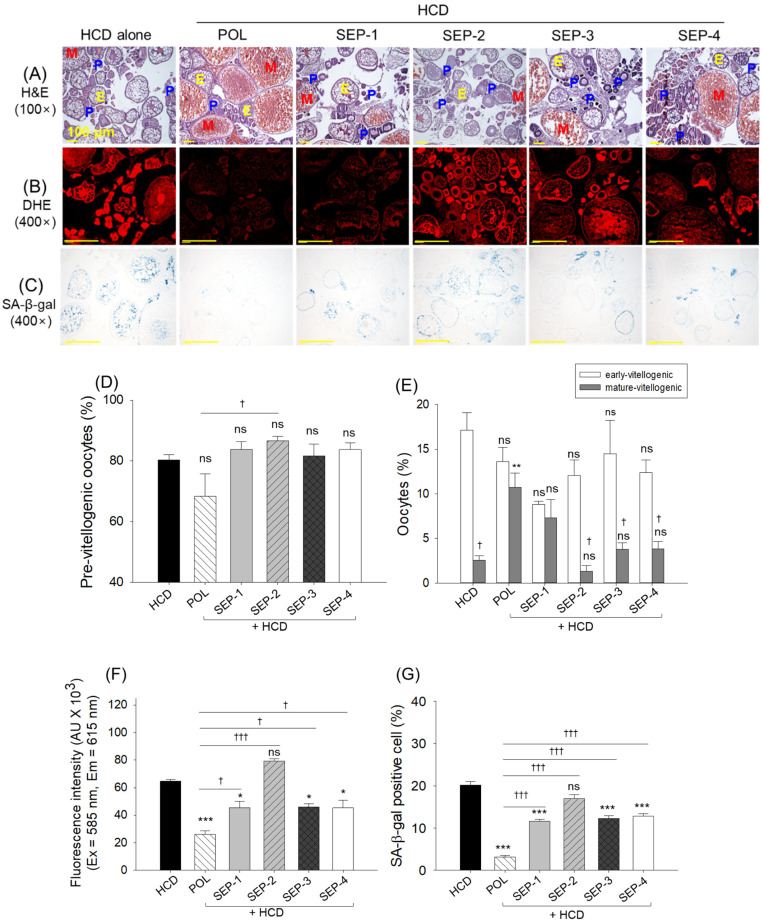
Histological analysis of the ovary from the hyperlipidemic zebrafish post 14 weeks of dietary intake of policosanol and different sugarcane extract powder. (**A**) Hematoxylin and eosin (H&E) staining (abbreviation represents P: pre-vitellogenic oocytes, E: early-vitellogenic oocytes, and M: mature-vitellogenic oocytes). (**B**) Dihydroethidium (DHE) fluorescent staining. (**C**) Senescent-associated β-galactosidase (SA-β-gal) staining [scale bar, 100 μm]. Quantification of (**D**) pre-vitellogenic oocytes, (**E**) early and mature-vitellogenic oocytes, (**F**) DHE fluorescent intensity, and (**G**) SA-β-gal positive cells. The *, **, and *** represent the statistical difference at *p* < 0.05, *p* < 0.01, and *p* < 0.001 with respect to the HCD group, while ^†^ (*p* < 0.05), and ^†††^ (*p* < 0.001) highlight the statistical significance vs. the POL group. The “ns” represents the non-significant difference. The acronyms HCD: high-cholesterol diet, POL: policosanol, and SEP: sugarcane extract powder.

**Figure 12 ijms-26-09524-f012:**
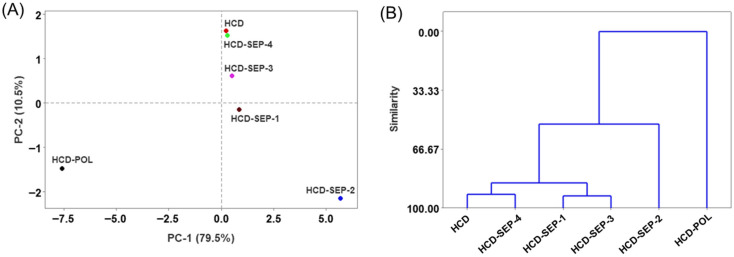
Multivariate analysis on the collective data from varied biochemical and histological analyses of zebrafish across the groups post 14 weeks of dietary intake. (**A**) Principal component analysis (PCA) loading plot, and (**B**) hierarchical cluster analysis (HCA). The acronyms HCD: high-cholesterol diet, HCD-POL: high-cholesterol diet with policosanol, while HCD-SEP-(1 to 4): high-cholesterol diet with different sugarcane extract powder.

**Table 1 ijms-26-09524-t001:** General information about policosanol (POL) and sugar cane extract powder (SEP) products.

Product Code	Product Manufacturer/Name and Country	Country of Origin (Source Material)	Ingredients	Policosanol Weight (mg)	Formulated Diet ^1^
POL	Raydel— Policosanol, Australia	Cuba	Policosanol-sugar cane wax alcohol (originated from Cuba), lactose mixed powder, crystal cellulose, hydroxypropylmethylcellulose, calcium carboxymethylcellulose, magnesium stearate, the garden blue pigment, titanium dioxide (color), glycerin fatty acid ester milk	18	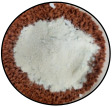
SEP-1	JW pharmaceutical, Republic of Korea	India	Red yeast powder, whole sugarcane extract powder (from India), bamboo water extract powder (Indian), green tea extract powder, seaweed calcium, broccoli powder, fructooligosaccharide, glycerin fatty acid ester	ND ^2^	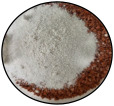
SEP-2	Nutricore, Republic of Korea	ND	Red Yeast Rice powder (88 mg), natto (soybean fermented powder), whole sugarcane extract powder, green tea powder, cotton seed powder, lemon extract powder, oat dietary fibers, fructooligosaccharide, C8MCT coconut oil powder, dried yeast	ND	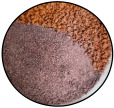
SEP-3	Esther formula, Republic of Korea	ND	Oligopowder, natto (soybean fermented powder), whole sugarcane extract powder, powdered green tea, spirulina powder, magnesium stearate, nicotine amide, glycerin fatty acid ester	ND	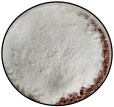
SEP-4	MayjuneNutri’s, Republic of Korea	ND	Isomaltooligosaccharide, whole sugarcane extract powder, seaweed calcium, corn protein extraction powder, magnesium stearate, oat dietary fiber, lemon peel extraction powder, garlic concentrated powder, dried yeast, glycerin fatty acid ester	ND	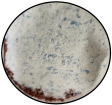

^1^ formulated diets: represent the supplementation of POL (1% *w*/*w*) or SEPs (1% *w*/*w*) with high-cholesterol diet (HCD); ^2^ ND: not disclosed.

## Data Availability

The data used to support the findings of this study are available from the corresponding author upon reasonable request.

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
