# Peer review of "Adverse Effect of Sugarcane Extract Powder (SEP) in Hyper-Lipidemic Zebrafish During a 14-Week Diet: A Comparative Analysis of Biochemical and Toxicological Efficacy Between Four SEPs and Genuine Policosanol (Raydel®)"

_ijms, 2025, doi:10.3390/ijms26199524_

Round 1
Reviewer 1 Report
Comments and Suggestions for Authors
Reviewer 1
General Perspective
The manuscript addresses an important comparative evaluation of sugarcane extract powders (SEPs) and authentic policosanol (POL) using a hyperlipidemic zebrafish model. The study is relevant and timely, but certain aspects of the manuscript could be further refined to enhance clarity, scientific rigor, and overall readability. Below I provide specific, section-wise suggestions.
Response: Thank you for your appreciation and invaluable suggestions.
- Abstract
I recommend briefly including a description of the biological activities of both policosanol and SEP prior to stating the study’s aim.
Response: Thank you for your insightful comments. Following the reviewer’s suggestion, changes have been made in the abstract section. Please refer to the text below and in the revised manuscript (lines 11-15).
“Sugarcane wax-derived policosanol (POL) is well recognized for its multifaceted biological activities, particularly in dyslipidemia management. Whereas sugar cane extract powder (SEP), prepared from whole sugar juice blended with supplementary components, has not been thoroughly investigated for its biological activities and potential toxicities. Herein, the comparative dietary effect of four distinct SEPs (SEP-1 to SEP-4) and Cuban sugarcane wax extracted POL were examined to prevent the pathological events in high-cholesterol diet (HCD)-induced hyperlipidemic zebrafish. Among the SEPs, a 14-week intake of SEP-2 emerged with the least zebrafish survival probability (0.75, log-rank: χ2=14.1, p=0.015), while the POL supplemented group showed the utmost survival probability. A significant change in body weight and morphometric parameters was observed in the SEP-2 supplemented group compared to the HCD group, while non-significant changes had appeared in POL, SEP-1, SEP-3, and SEP-4 supplemented groups. The HCD elevated total cholesterol (TC) and triglyceride (TG) levels were significantly minimized by the supplementation of POL, SEP-1, and SEP-2. However, an augmented HDL-C level was only noticed in POL-supplemented zebrafish. Likewise, only the POL-supplemented group showed a reduction in blood glucose, malondialdehyde (MDA), AST, and ALT levels, and an elevation in sulfhydryl content, paraoxonase (PON), and ferric ion reduction (FRA) activity. Also, plasma from the POL-supplemented group showed the highest antioxidant activity and protected zebrafish embryos from carboxymethyllysine (CML)-induced toxicity and developmental deformities. POL effectively mitigated HCD-triggered hepatic neutrophil infiltration, steatosis, and the production of interleukin (IL)-6 and inhibited cellular senescence in the kidney and minimized the ROS generation and apoptosis in the brain. Additionally, POL substantially elevated spermatozoa count in the testis and safeguarded ovaries from HCD-generated ROS and senescence. The SEP products (SEP-1, SEP-3, and SEP-4) showed almost non-significant protective effect; however, SEP-2 exhibited an additive effect on the adversity posed by HCD in various organs and biochemical parameters. The multivariate examination, employing principal component analysis (PCA) and hierarchical cluster analysis (HCA), demonstrates the positive impact of POL on the HCD-induced pathological events in zebrafish, which are notably diverse, with the effect mediated by SEPs. The comparative study concludes that POL has a functional superiority over SEPs in mitigating adverse events in hyperlipidemic zebrafish.”
- Suggestion: remove Paraoxonase from the Keywords. The broader context of the study is already clear with the other terms.
Response: Thank you for your insightful comments. The word “Paraoxonase” has now been removed from the keyword section. Please refer to the text below and in the revised manuscript (lines 39).
Keywords: dyslipidemia; fatty liver; oxidative stress; policosanol; senescence; spermatozoa
- Introduction
While the genomic and metabolic similarities between zebrafish and humans are well described, it remains unclear why the study specifically measured MDA, AST, ALT, sulfhydryl content, paraoxonase (PON), and ferric ion reduction (FRA) activity, alongside the other biomarkers employed.
I suggest providing a more detailed explanation of the rationale for selecting these biomarkers and enzymes, as well as their role as indicators of oxidative stress in different organs. Adding more references in this regard would strengthen the introduction.
Response: Thank you for your insightful comments. In the introduction section, the rationale behind the selection of the biochemical markers used and their role as indicators of oxidative stress and disease, supported by relevant references, has now been included. Please refer to the text below and in the revised manuscript (lines 88-102). Hope this will be acceptable to the honorable reviewer. Thank you!
“In view of this, the present study aimed to comparatively evaluate the 14-week dietary intake of four distinct SEP products and policosanol (POL) on the survivability, blood lipoprotein profile, and histopathological changes in vital organs, including liver, kidney, brain, testes, and ovaries, of hyperlipidemic zebrafish. Furthermore, the study investigated the effect of SEPs and POL on plasma oxidative stress and antioxidant variables, such as malondialdehyde (MDA), sulfhydryl groups, ferric ion reduction ability (FRA), and paraoxonase (PON) activity, along with hepatic function biomarkers aspartate aminotransferase (AST) and alanine aminotransferase (ALT). The selection of these markers was based on the fact that MDA represents a key lipid peroxidation product and is widely recognized as an essential indicator of oxidative stress [15], whereas FRA reflects the overall antioxidant capacity of blood [16]. The sulfhydryl groups act as primary antioxidants that scavenge peroxyl radicals [17,18], and their decreased level is linked to diseases like rheumatoid arthritis, coronary heart disease, and kidney disorders [19,20]. PON, an HDL-associated enzyme, exhibits an inverse relationship with lipid peroxidation and oxidative stress [21], and its decreased level is associated with myocardial infarction [22] and chronic liver disease [23]. AST and ALT serve as key biomarkers of liver function, with elevated levels indicating poor liver health.”
- Materials and Methods
It would be more appropriate to present the “Supplementary Table S1: Composition of policosanol (POL) and sugarcane extract powder (SEP)” in place of “Table 1. General information about POL and SEP”, so that comprehensive compositional data are available in the main text.
Response: Thank you for your insightful comments and keen observation. Following the reviewer’s suggestion, Supplementary Table S1 has now been moved to the manuscript materials and methods section, 4.1. Please refer to the text and table enclosed below and in the revised manuscript (Section 4.1, line 700, and Table 1).
4.1. Materials
Policosanol (abbreviated as POL batch no: 2325), a mixture of eight long aliphatic alcohols (LCAA, C24 to C34), was extracted from sugarcane wax sourced from Cuba, and was complementary provided by the Raydel® Pty. Ltd., Thornleigh, NSW, Australia. Four different sugarcane extract powders (SEP), namely JW Pharmaceuticals, Gwacheon-si, Republic of Korea (abbreviated as: SEP-1, batch: 1183001), Nutricore, Suwon, Republic of Korea (abbreviated as: SEP-2, batch no: 2400029), Esther formula, Seoul, Republic of Korea (abbreviated as: SEP-3, batch no: 2400561), and MayjuneNutri’s, Seoul, Republic of Korea (abbreviated as: SEP-4, batch no: 2983411), were purchased from the local market, Daegu, Republic of Korea. A specification of POL and SEP-1 to SEP-4 is provided in Table 1. All the other chemicals and reagents were used as supplied unless otherwise stated.
Table 1. General information about policosanol (POL) and sugar cane extract powder (SEP) products.
- In “Zebrafish culturing”, please clarify the rationale for using mixed sexes (male and female) instead of separating them. Could this choice have behavioral, hormonal, or territorial implications that might bias the results?
Response: Thank you for your insightful comments. Humbly, we would like to clarify that while forming the groups (I-VI), zebrafish of mixed sexes (male and female) were separated, with each group (n = 28) consisting of 14 males and 14 females. In the previously submitted (non-revised) manuscript, we had provided the individual mortality data of male and females, along with their respective MDA levels, as supplementary Figure S1 and S2. The rational for including both male and female zebrafish was to evaluate the effects of SEPs and POL on the testis and ovary, respectively. Now in the revised Materials and Method section (section 4.4), the grouping procedure is clearly explained, indicating the separation of sexes during the experimental setup to avoid any ambiguity. Please refer to the text provided below and in the revised manuscript (section 4.4, lines 729-736).
4.4. Feeding of different diets to zebrafish
A total of 168 zebrafish were allocated into six cohorts (n=28/group), with each group comprising an equal number of male (n=14) and female (n=14) zebrafish that were kept in separate tanks. The groups were assigned the following diets: HCD alone (group I), HCD+POL (group II), HCD+SEP-1 (group III), HCD+SEP-2 (group IV), HCD+SEP-3 (group V), and HCD+SEP-4 (group VI). Within each group, zebrafish (n=28, 14 male and 14 female) were further distributed into 4 separate tanks (i.e. n=7/tank, 2 tanks containing male and 2 tanks containing female) and provided with the specified diet of 70 mg/tank two times in the day (~9am and 6pm), amounting to a total of 140 mg/tank/day (equivalent to ~20 mg/zebrafish/day). Prior to the specified dietary intervention, all zebrafish in groups I-VI were pre-fed exclusively with HCD for 7 weeks to induce metabolic stress (hyperlipidemic zebrafish), after which they were maintained on their designated diets. Notably, at the end of 7 weeks of pre-feeding, blood was collected from ten randomly selected zebrafish and analyzed for biochemical analysis to ensure the induction of metabolic stress. Supplementary Table S1 depicts the change in the blood-biochemical profile after 7 weeks of pre-feeding with HCD compared to their initial blood-biochemical profile while consuming the ND diet.
Food consumption among all groups was examined using the previously described method for zebrafish [72]. The food consumption was determined at the beginning (week 0) and at the end of 14 weeks using the formula: [(Total food given per group – residual food per group)/total food given per group] × 100. The residual food was quantified post 30 min of the feed supplied.
6.In “Formulation of different diets”, please clarify whether the diets were freshly prepared each day and how any excess was stored (temperature and conditions). More details would improve reproducibility.
Response: Thank you for your insightful comments and keen observation. Different formulated diets were prepared at the beginning of the study, and the diets were stored in a refrigerator at 4°C. After use for the feeding purpose (during 14 weeks), diets were always kept in the refrigerator. A concerning statement has now been included in the revised manuscript. Please refer to the text below and in the revised manuscript (section 4.3, lines 716-727). Thank you!
4.3. Formulation of different diets
Normal tetrabit (ND) was amalgamated with cholesterol (4% w/w) to make the high cholesterol diet (HCD). In brief, 750 g of ND was suspended in 30 g of cholesterol (i.e., final 4% w/w) and mixed thoroughly using a spatula. Subsequently, chloroform was added, and the mixture was vigorously agitated to distribute the cholesterol evenly. Finally, the chloroform was evaporated entirely in the fume hood, and the HCD was further used to prepare different POL and SEPs formulated diets. Briefly, HCD (100 g) was mixed either with 1% (w/w) POL or SEP-1/ SEP-2/ SEP-3/ SEP-4 to make five different dietary formulations named HCD+POL, HCD+SEP-1/ SEP-2/ SEP-3 and SEP-4, respectively (as depicted in Table 1). Different diets, i.e., HCD and HCD supplemented either with POL or SEPs, were preserved in a refrigerator (4°C) and used to feed the zebrafish for 14 weeks. After using it for feeding (every day), different diets were transferred to the refrigerator.
- Regarding Food consumption, the formula used resembles an ad libitum calculation. Please specify whether this equation is standardized for aquatic models or adapted from terrestrial animal studies.
Response: Thank you for your insightful comments. Yes, we have standardized the method for studying zebrafish feed consumption. A statement with the supporting reference has now been included in the revised manuscript. Kindly refer to the text below and in the revised manuscript (lines 739-750). Thank you!
4.4. Feeding of different diets to zebrafish
A total of 168 zebrafish were allocated into six cohorts (n=28/group), with each group comprising an equal number of male (n=14) and female (n=14) zebrafish that were kept in separate tanks. The groups were assigned the following diets: HCD alone (group I), HCD+POL (group II), HCD+SEP-1 (group III), HCD+SEP-2 (group IV), HCD+SEP-3 (group V), and HCD+SEP-4 (group VI). Within each group, zebrafish (n=28, 14 male and 14 female) were further distributed into 4 separate tanks (i.e. n=7/tank, 2 tanks containing male and 2 tanks containing female) and provided with the specified diet of 70 mg/tank two times in the day (~9am and 6pm), amounting to a total of 140 mg/tank/day (equivalent to ~20 mg/zebrafish/day). Prior to the specified dietary intervention, all zebrafish in groups I-VI were pre-fed exclusively with HCD for 7 weeks to induce metabolic stress (hyperlipidemic zebrafish), after which they were maintained on their designated diets. Notably, at the end of 7 weeks of pre-feeding, blood was collected from ten randomly selected zebrafish and analyzed for biochemical analysis to ensure the induction of metabolic stress. Supplementary Table S1 depicts the change in the blood-biochemical profile after 7 weeks of pre-feeding with HCD compared to their initial blood-biochemical profile while consuming the ND diet.
Food consumption among all groups was examined using the previously described method for zebrafish [72]. The food consumption was determined at the beginning (week 0) and at the end of 14 weeks using the formula: [(Total food given per group – residual food per group)/total food given per group] × 100. The residual food was quantified post 30 min of the feed supplied.
Supporting reference:
- Cho, K.-H.; Lee, Y.; Bahuguna, A.; Lee, S.H.; Yang, C.-E.; Kim, J.-E.; Kwon, H.-S. The consumption of beeswax alcohol (BWA, Raydel®) improved zebrafish motion and swimming endurance by protecting the brain and liver from oxidative stress induced by 24 weeks of supplementation with high-cholesterol and D-galactose diets: A comparative analysis between BWA and coenzyme Q10. Antioxidants2024, 13, 1488.
- In “Blood biochemical analysis”, references are missing for the methods used to quantify sulfhydryl content and paraoxonase (PON) activity.
Response: Thank you for your insightful comments and keen observations. A relevant reference for the determination of sulfhydryl content and paraoxonase (PON) activity has now been included in the revised manuscript. Please refer to the text below and in the revised manuscript (lines 777-778). Thank you!
Plasma sulfhydryl content and paraoxonase (PON) activity were determined using the earlier described method [73]. For the quantification of sulfhydryl content, an equal volume of plasma (50 μL, 1 mg/mL protein) was mixed with 0.4% of 5,5-dithio-bis-(2-nitrobenzoic acid) and content was incubated at room temperature (RT) for 2 h. Subsequently, absorbance at 412 nm was recorded, and the sulfhydryl content was quantified as mmol/mg protein using the molar absorbance coefficient (ε) 1.36×104M–1cm–1 of formed product (5-thiol-2-nitrobenzoic acid).
Supporting reference:
Cho, K.-H.; Kim, J.-E.; Lee, M.-S.; Bahuguna, A. Oral supplementation of ozonated sunflower oil augments plasma antioxidant and anti-inflammatory abilities with enhancement of high-density lipoproteins functionality in rats. Antioxidants 2024, 13, 529.
- In “In vivo functionality evaluation of the plasma”, please cite the methodological reference employed.
Response: Thank you for your insightful comments and keen observation. The relevant reference in support of the “in vivo functionality evaluation of plasma” has now been included in the respective section. Please refer to the text below and in the revised manuscript lines (section 4.8, line 793). Thank you!
4.8. In vivo functionality evaluation of the plasma
The functionality of the plasma obtained from the different groups (I-VI) was analyzed in zebrafish embryos against carboxymethyllysine (CML) triggered toxicity, using an earlier described method [73].
Supporting reference:
Cho, K.-H.; Kim, J.-E.; Lee, M.-S.; Bahuguna, A. Oral supplementation of ozonated sunflower oil augments plasma antioxidant and anti-inflammatory abilities with enhancement of high-density lipoproteins functionality in rats. Antioxidants 2024, 13, 529.
- For the Histological analysis, it should be clarified whether all sections were taken from the same anatomical region of each organ, or if there was variability in sampling sites. This would help in better interpreting histological comparisons.
Response: Thank you for your insightful comments and invaluable suggestions. For the histological analysis, the whole organs from different groups were fixed in FSC22 frozen solution in the same orientation to prepare a solid block, ensuring that nearly similar anatomical regions were oriented for sectioning. A concerning statement has now been included in the revised manuscript to avoid ambiguity. Please refer to the text below and in the revised manuscript lines (section 4.9, lines 821-829). Thank you!
4.9. Histological analysis
For the histological analysis, the whole organ (liver, kidney, brain, testis, and ovaries) was blocked in the FSC22 frozen solution (Leica, Nussloch, Germany). Notably, the respective organs from different groups were inserted in the FSC22 frozen solution in the same orientation to prepare the solid block, orienting nearly similar anatomical regions for sectioning. The fixed tissue block was sectioned (7 μm thick) using the cryo-microtome (Leica CM-1510S, Nussloch, Germany). The sections obtained from the different tissues were processed for the hematoxylin and eosin (H&E) staining using the previously described method [76].
- Results and Discussion
In Figure 2 (Organ morphology and organ weight), the Y-axis labels should preferably be expressed as “mg/b.w.” (body weight) rather than “mg/zebrafish” or “mg/rat”.
Response: Thank you for your insightful comments. As recommended by the reviewer, the labels of the Y-axis have been modified. Please refer to the image below and in the revised manuscript, Figure 2. Thank you!
Revised Figure 2
- To better contextualize the findings, a normal control group (standard diet) should be included, or at least baseline values (body weight, body length, organ weights, biochemical parameters, etc.) for the zebrafish strain should be mentioned. Since the HCD diet itself appears to alter these parameters, adding a reference table or description of normal values would be highly informative.
Response: Thank you for your insightful comments and valuable suggestions. In the revised manuscript, we have now included the baseline values (week 0) for body weight, total cholesterol, triglycerides, high-density lipoprotein cholesterol, and plasma oxidative stress and antioxidant markers. For easy reference, these values are now provided in tabular form as a supplementary Table S1. Additionally, for comparative context, we have also included the corresponding values from zebrafish maintained on a normal diet (ND) during the 2-week acclimatization period, alongside the HCD-pre-feeding base values (7 weeks of pre-feeding with HCD). Please refer to the text below and in the revised manuscript (lines 144-148, 200-218, 242-272) and supplementary Table S1. Thank you!
2.1. Survivability and alteration in body weight
The repeated ANOVA outcomes utilizing the multivariate tests (Pillai’s Trace, Wilks Lambda, Hotelling’s Trace, and Roy’s largest root) revealed a significant effect of time (0-14 weeks, F=148, p=0.00) on the elevation of body weight among all the groups (Figure 1 B). Interestingly, all the groups except SEP-2 exhibited nearly identical magnitudes of body weight changes at specific time points that corresponded to the body weight changes observed in the HCD (control) group at the respective time. In contrast, the SEP-2 supplemented group displayed the least body weight gain compared to other groups. Compared to the initial day (week 0) a notable ~2-fold enhancement of the body weight was observed after 14 weeks in the HCD, POL, SEP-1, SEP-3, and SEP-4 supplemented groups. In contrast, after 14 weeks, only a 1.7-fold body weight enhancement with respect to the initial day (week 0) was noticed in the SEP-2 supplemented group (Figure 1 B). Notably, at 14-week SEP-2 supplemented group attained a significantly 18% (p<0.05) lower body weight (506.8±29.8 mg) than the body weight of the HCD consumed group (614.8±27.5 mg), whereas no significant changes in the body weight compared to the HCD group were noticed in rest of the groups (i.e. POL, SEP-1, SEP-3, and SEP-4). Consistently, morphometric analysis revealed a significant change in the body length (BL)/body depth (BD) ratio in the zebrafish from the SEP-2 supplemented groups compared to the HCD group (Figure 1 D). Unlike this, non-significant changes in the BL/BD ratio were noticed in POL, SEP-1, SEP-3, and SEP-4 groups with respect to the HCD group.
2.3. Blood lipoprotein profile and glucose levels
An elevated total cholesterol (TC, 198.5±1.5 mg/dL) level was noticed in the HCD-supplemented group (Figure 3 A), which is 1.2-fold higher than the basal level (168.3±12.6 mg/dL) detected at week 0 (supplementary Table S1). The co-supplementation of POL (138.1±1.3 mg/dL), SEP-1 (167.4±2.5mg/dL), and SEP-2 (175.1±1.4 mg/dL) effectively reduced the HCD elevated TC level (Figure 3 A). However, when compared with SEP-1 and SEP-2, POL displayed significantly lower TC levels of 18% and 12%, reflecting POL's higher efficacy over SEP-1 and SEP-2. In contrast, no significant effect of SEP-3 and SEP-4 supplementation was noticed on the HCD-induced TC levels.
Like TC, the basal (week 0) triglycerides (TG, 90.6±6.3 mg/dL) level (supplementary Table S1) was 18% enhanced (106.6±0.4 mg/dL) in the HCD supplemented group which was significantly reduced following the co-supplementation of POL (70.6±2.3 mg/dL), SEP-1 (86.6±3.8 mg/dL), and SEP-2 (88.2±2.6 mg/dL) (Figure 3 B). Nonetheless, the POL-supplemented group displayed ~1.2-fold higher efficacy than the SEP-1 and SEP-2-supplemented groups to reducing the HCD-induced TG levels. A non-significant effect of SEP-3 and Sep-4 was noticed to reduce the HCD elevated TG level.
The week 0 high-density lipoprotein cholesterol (HDL-C) level (51.5±3.4 mg/dL) (supplementary Table S1) was substantially reduced up to 45.9±1.9 mg/dL following 14-week supplementation of HCD. The HCD-depleted HDL-C level was significantly 32.4% enhanced following the co-supplementation of POL (60.8±1.0 mg/dL) (Figure 3 C).
2.4. Oxidative variables, antioxidant abilities and hepatic function biomarkers of blood
A week 0 plasma malondialdehyde (MDA) level (8.2±0.4 μM) (supplementary Table S1) increased up to 10.5±0.5 μM in response to HCD-supplementation for 14 weeks. The HCD elevated MDA level was significantly reduced by 21% following the co-supplementation of POL (8.1±0.3 μM) (Figure 4 A). Unlike POL, a non-significant effect of SEPs supplementation was noticed on the HCD elevated plasma MDA level. However, SEP-2 exacerbates the HCD-induced MDA level, as evidenced by a 32% higher MDA level in the SEP-2 supplemented group (13.8±0.8 μM) compared to the HCD group.
A substantially compromised plasma sulfhydryl content (11.7±0.5 mmol/mg protein) (Figure 4 B) was observed post 14-week intake of HCD with respect to the week 0 level (12.2±0.5 mmol/mg protein) (supplementary Table S1). A significantly 22% higher plasma sulfhydryl content was quantified in the POL supplemented group (14.3±0.6 mmol/mg protein) compared to the HCD group (Figure 4 B). Unlike POL, no SEP products showed a significant augmentation of HCD diminished plasma sulfhydryl content. Notably, Sep-2 supplementation exerted a negative impact, resulting in a 23% decrease in plasma sulfhydryl content relative to the HCD group.
Compared to the initial (week 0) FRA (252.9±15.4 μM) and PON±0.3 activity (6.9 μU/L/min) (supplementary Table S1), a notable diminished FRA (187.8±5.5 μM) and PON activity (5.8±0.4 μU/L/min) was observed after 14-week intake of HCD (Figure 4 C, D). The HCD-compromised plasma FRA and PON activity were significantly 1.5-fold (285.4±8.6 μM) and 2-fold (11.6±0.5 μU/L/min) enhanced by the co-supplementation of POL (Figure 4 C, D). Nevertheless, a non-significant effect of SEP products was noticed on the elevation of plasma FRA and PON activity, which was compromised by the supplementation of HCD. Strikingly, the SEP-2 supplementation displayed a detrimental effect on the plasma FRA and PON activity as reflected by a significantly 1.2-fold and 1.5-fold reduced FRA and PON activity, respectively, relative to the HCD group.
HCD elevated plasma AST and ALT levels were significantly reduced by 29.9% and 26.7% following the co-supplementation of POL. In contrast, no significant (p>0.05) effect of any SEP products was noticed towards the reduction of HCD elevated AST and ALT levels (Figure 4 E, Furthermore, the SEP-2 supplementation demonstrated a significant 1.2-fold (p<0.05) additive effect toward the HCD elevated AST level.
Supplementary Table S1
Supplementary Table S1: A comparative blood biochemical profile of zebrafish maintained on the normal diet (ND) that subsequently transferred to the high-cholesterol diet (HCD) for 7 weeks (pre-feeding).
|
Plasma biochemical parameters |
Zebrafish |
|
|
ND (14 weeks aged) |
HCD (7-week consumption) (pre-feeding) |
|
|
Total cholesterol (TC) |
130.7 ± 2.1 mg/dL |
168.3 ± 12.6 mg/dL |
|
Triglycerides (TG) |
60.1 ± 1.4 mg/dL |
90.6 ± 6.3 mg/dL |
|
High density lipoprotein cholesterol (HDL-C) |
70.4 ± 4.4 mg/dL |
51.5 ± 3.4 mg/dL |
|
Malondialdehyde (MDA) |
4.97 ± 0.2 μM |
8.2 ± 0.4 μM |
|
Sulfhydryl content |
15.2 ± 0.3 mmol/mg |
12.2 ± 0.5 mmol/mg |
|
Ferric reducing activity (FRA) |
454.8 ± 10.6 μM |
252.9 ± 15.4 μM |
|
Paraoxonase activity (PON) |
12.1 ± 0.2 μU/L/min |
6.9 ± 0.3 μU/L/min |
- Please revise grammar and punctuation carefully throughout the discussion, as there are minor spelling and formatting inconsistencies (e.g., highlighted words in red, punctuation issues).
Response: Thank you for your insightful comments. The manuscript has been thoroughly evaluated to address typographical and formatting-related issues. Thank you!
- In the HCD + SEP-2 group, could the observed high toxicity be attributed to a specific compound in red yeast rice extract, or is it related to its pro-oxidant activity? This distinction would strengthen the interpretation.
Response: Thank you for your insightful comments and keen observation. As the exact composition of the SEP-2 product is not described, it’s tough to say whether the toxicity is imparted by any specific compound in red yeast rice. However, red yeast rice has been recognized to contain various toxic compounds that lead to severe damage. Furthermore, it is evident from the present findings (blood and histological analysis) that SEP-2 has a strong pro-oxidant activity that leads to several detrimental effects. A statement concerning this has now been included in the discussion section. Kindly refer to the text below and in the revised manuscript (lines 561-564, 618-620). Thank you!
“After 14 weeks of supplementation, the survivability of zebrafish in the SEP-2 supplemented group was drastically reduced, indicating a significant adverse effect of SEP-2 on zebrafish survival. Unlike this, better survivability was observed in the other SEP-supplemented groups. Nevertheless, the most notable effect with no zebrafish mortality was observed in the POL-supplemented group, underscoring the safe nature of POL over SEP products, precisely with respect to SEP-2. The high presence of red yest rice extract (RYR) in the SEP-2 (Table 1) is the primary reason for the high toxicity of SEP-2 towards zebrafish, as RYR has been recognized to be contaminated with a variety of toxic substances that pose a serious health concern [31]. The statement aligns with earlier reports that highlight the severe adverse consequences of RYR consumption [31,32].”
“MDA is a vital stress marker [47] whose higher levels indicate lipid peroxidation. Likewise, plasma sulfhydryl levels also indicate a stressful environment, and their diminished levels are associated with various diseases [19,20,48]. A positive effect of POL supplementation was noticed towards the HCD-disturbed plasma MDA and sulfhydryl content. Likewise, POL supplementation effectively improved plasma PON and FRA activity, indicating a positive effect of POL on enhancing plasma antioxidant status. The results aligned with previous reports describing the modulatory effect of POL on plasma MDA [44,49], FRA, and PON [9] activity compromised by external stress. Contrary to POL, no SEP products positively modulate the HCD-disturbed plasma MDA, sulfhydryl content, FRA, and PON activity. Even more in response to SEP-2, the plasma oxidative stress and antioxidant parameters are severely compromised, which were significantly worse than the effect observed in the HCD group, indicating a strong pro-oxidant effect of SEP-2.”
- Conclusions
It is important to clearly emphasize the limitations of this study in the conclusion section.
Response: Thank you for your insightful comments. The limitation of the study has now been included in the conclusion section of the manuscript. Please refer to the text below and in the revised manuscript (lines 870-872). Thank you!
- Conclusions
A 14-week dietary intervention study showed a severe adverse effect of SEP-2 (containing RYR) on the survivability of hyperlipidemic zebrafish, while no visible adverse effect was noted for POL. In contrast to the POL, SEP-2 elevated oxidative stress and diminished antioxidant variables of plasma. Similarly, SEP-2 exacerbates the HCD-induced fatty liver, hepatic inflammation, damage to the kidney, brain, and reproductive organs, while POL displays a notable protective effect against the HCD-induced adverse events. The other SEP products (SEP-1, SEP-3, and SEP-4) displayed no improved effect against HCD-induced dyslipidemia. The antioxidant and anti-inflammatory effects of POL contributed to its protective effect against HCD-induced dyslipidemia and oxidative stress. Whereas SEP-2 displayed a pro-oxidant effect, consequently aggravating the HCD-induced adverse events, leading to severe toxicity. The study concludes the functional divergence between POL and SEP products and underscores the specific role of POL in alleviating HCD-mediated adverse outcomes. Nevertheless, a limited number of zebrafish per group and the unavailability of exact compositional details of SEPs emerged as a fundamental limitation of the study that needs to be addressed in the future.
Reviewer 2 Report
Comments and Suggestions for Authors
Reviewer 2
- The topic is original and relevant to the field and the results and methods are presented adequately to the thesis. The references were appropriate, up to date, no self-citing were not discovered.
Response: Thank you for your appreciation and invaluable suggestions.
- While reading the manuscript, I noticed occasional minor writing and punctuation errors, etc. The text should be reviewed for this purpose.
Response: Thank you for your insightful comments. The manuscript has been thoroughly evaluated to address typographical and formatting-related issues. Thank you!
- in 'introduction' the section on plant origin should be expanded to include information about the plant itself.
Response: Thank you for your insightful comments. A brief introduction about plant origin and its industrial applicability has now been added in the introduction section. Please refer to the text below and in the revised manuscript (lines 50-56). Thank you!
Policosanol is a generic name used for the mixture of long-chain aliphatic alcohols (LCAAs) [1] that can be extracted from a variety of sources, including plants (e.g., wheat germ, sugarcane, maize, grapes, rice bran) and animals (e.g., bees) [2,3]. However, policosanol was first extracted from the sugarcane (Saccharum officinarum L.) wax originated from Cuba in the early 90s that harbor a unique mixture of eight LCAAs namely tetracosanol (C24, 0.1–2%), hexacosanol (C26, 3–10%), heptacosanol (C27, 0.1–3%), octacosanol (C28, 60–70%), nonacosanol (C29, 0.1–2%), triacontanol (C32, 10–15%) and tetratriacontanol (C34, 0.1–5%) by the Centro Nacional de Investigaciones Científicas (CNIC, Havana, Cuba) [1]. Sugarcane is a perennial tropical grass of the Poaceae (Gramineae) family and is mainly cultivated in arid and semi-arid regions for sugar production. Beyond sugar, it serves as a valuable raw material to produce ethanol, vinegar, and wine [4]. Importantly, sugarcane processing yields several byproducts such as bagasse, a lignin-rich residue, and press mud, which is utilized to extract sugarcane wax, a precursor for the extraction of policosanol [5]. Owing to these versatile applications, sugarcane and its derivatives hold significant industrial importance, including in the cosmetic and pharmaceutical sectors [5,6]. Policosanol has been recognized for its diverse functionality, including therapeutic impact on hypertension, intermittent claudication, ischemic heart disease, and Alzheimer's disease [2,7]. In addition, the inhibitory effect of policosanol has been documented for the synthesis of platelet-aggregating thromboxane B2 (TXB2), consequently attenuating platelet aggregation [7].
- Does the choice of research model, which is zebrafish, affect the experiment itself? Have similar studies been conducted on other species, e.g., mice?
Response: Thank you for your insightful comments. The choice of the zebrafish as a model organism was based on its high genomic resemblance to humans and close similarities with human lipoprotein metabolism. Furthermore, HCD-induced responses in zebrafish are parallel to many human pathophysiological conditions, making it a suitable model for preclinical research related to human liver, kidney, and reproductive disorders. Based on these features, there is a high chance that results obtained in the zebrafish can be translated into humans in a somewhat similar manner. However, it must be noted that despite several similarities, zebrafish also differ in many physiological aspects from humans; therefore, results obtained from the zebrafish must be examined prudently with respect to their applicability to humans.
Furthermore, studies conducted on mice and rats are available for policosanol (POL); however, to the best of our knowledge, no such studies are available for sugarcane extract powder (SEP). A concerning statement has now been included in the revised manuscript. Please refer to the text below and in the revised manuscript (lines 594-600, 637-639). Thank you!
As a primary mechanism, the inhibitory effect of POL on CETP activity has been recognized as crucial in the elevation of HDL-C levels [44]. The current findings align with earlier reports documenting the positive effect of POL in counteracting dyslipidemia in mice and rats [2]. However, no study has demonstrated the effect of SEP on animal models other than zebrafish. The lipid-lowering effect of POL in physiologically distinct species, such as mammals (mice and rats) and vertebrates (zebrafish), testifies to the broad applicability of POL in managing dyslipidemia.
HCD and dyslipidemia adversely affect the liver, provoking hepatic inflammation [52,53]. Likewise, high neutrophil counts, fatty liver, and IL-6 production were observed in the HCD group, which was significantly protected by the consumption of POL. The results are in good agreement with established literature, which documents the hepatoprotective role of POL tested in models such as rats [54] and zebrafish [55].
- The author explained the use of a given animal model in the context of lipid disorders, but is there information on other aspects in other species?
Response: Thank you for your insightful comments. Following the reviewer recommendations, now in the revised manuscript we have included aspects other than lipid metabolism, for which zebrafish served as good animal model. Please refer to the text below and in the revised manuscript (lines 103-116). Thank you!
Zebrafish were chosen as the experimental model due to their high genomic resemblance to humans [24], and more specifically, the close similarity of their lipid metabolism pathways. Zebrafish harbor many key lipid metabolism enzymes, receptors, and lipoproteins analogous to those in humans, including cholesteryl ester transfer protein (CETP), a critical component of human lipid metabolism that is absent in mice [25], rendering zebrafish an ideal model organism for lipoprotein research [26]. Moreover, the high-cholesterol diet (HCD) induced events in zebrafish mimicked with human hepatic steatosis and exhibited gene expression patterns comparable to those observed in mammalian models [27]. Furthermore, HCD-induced responses in zebrafish are parallel to many human pathophysiological conditions, including macrophage lipid accumulation and vascular lesion formation [27]. In addition, zebrafish have proven to be a good model for preclinical research related to human liver [28], kidney [29], and reproductive ailments [30]. Consequently, compounds showing efficacy in zebrafish have a high likelihood of exhibiting comparable responses in human clinical studies.
- A graphical abstract would be also beneficial.
Response: Thank you for your insightful comments. A graphical abstract has now been included in the revised manuscript. Please refer to the image below and the graphical abstract section of the manuscript. Thank you!
Graphical abstract
- The 'conclusions' or 'discussion' section should have some 'summing up' part in it. The conclusions are not quite visible in that section, giving all the data provided in the main text.
Response: Thank you for your insightful comments. In the conclusion section, a brief summing up part has now been included. Please refer to the text below and in the revised manuscript, section 5 (Lines 858-872). Thank you!
- Conclusions
A 14-week dietary intervention study showed a severe adverse effect of SEP-2 (containing RYR) on the survivability of hyperlipidemic zebrafish, while no visible adverse effect was noted for POL. In contrast to the POL, SEP-2 elevated oxidative stress and diminished antioxidant variables of plasma. Similarly, SEP-2 exacerbates the HCD-induced fatty liver, hepatic inflammation, damage to the kidney, brain, and reproductive organs, while POL displays a notable protective effect against the HCD-induced adverse events. The other SEP products (SEP-1, SEP-3, and SEP-4) displayed no improved effect against HCD-induced dyslipidemia. The antioxidant and anti-inflammatory effects of POL contributed to its protective effect against HCD-induced dyslipidemia and oxidative stress. Whereas SEP-2 displayed a pro-oxidant effect, consequently aggravating the HCD-induced adverse events, leading to severe toxicity. The study concludes the functional divergence between POL and SEP products and underscores the specific role of POL in alleviating HCD-mediated adverse outcomes. Nevertheless, a limited number of zebrafish per group and the unavailability of exact compositional details of SEPs emerged as a fundamental limitation of the study that needs to be addressed in the future.
Round 2
Reviewer 1 Report
Comments and Suggestions for Authors
I would like to inform you that I have no further comments regarding your manuscript. I sincerely appreciate the time and effort you have dedicated to addressing the observations provided.